# Mechanism of cargo-directed Atg8 conjugation during selective autophagy

Dorotea Fracchiolla[1†], Justyna Sawa-Makarska[1†], Bettina Zens[1], Anita de Ruiter[2‡], Gabriele Zaffagnini[1], Andrea Brezovich[1], Julia Romanov[1], Kathrin Runggatscher[1], Claudine Kraft[1], Bojan Zagrovic[2], Sascha Martens[1*]

[1]Department of Biochemistry and Cell Biology, Max F. Perutz Laboratories (MFPL), University of Vienna, Vienna Biocenter (VBC), Vienna, Austria; [2]Department of Structural and Computational Biology, Max F. Perutz Laboratories (MFPL), University of Vienna, Vienna Biocenter (VBC), Vienna, Austria

*For correspondence: sascha.martens@univie.ac.at

[†]These authors contributed equally to this work

Present address: [‡]Institute for Molecular Modelling and Simulation, University of Natural Resources and Life Sciences, Vienna, Austria

Competing interests: The authors declare that no competing interests exist.

**Abstract** Selective autophagy is mediated by cargo receptors that link the cargo to the isolation membrane via interactions with Atg8 proteins. Atg8 proteins are localized to the membrane in an ubiquitin-like conjugation reaction, but how this conjugation is coupled to the presence of the cargo is unclear. Here we show that the *S. cerevisiae* Atg19, Atg34 and the human p62, Optineurin and NDP52 cargo receptors interact with the E3-like enzyme Atg12~Atg5-Atg16, which stimulates Atg8 conjugation. The interaction of Atg19 with the Atg12~Atg5-Atg16 complex is mediated by its Atg8-interacting motifs (AIMs). We identify the AIM-binding sites in the Atg5 subunit and mutation of these sites impairs selective autophagy. In a reconstituted system the recruitment of the E3 to the prApe1 cargo is sufficient to drive accumulation of conjugated Atg8 at the cargo. The interaction of the Atg12~Atg5-Atg16 complex and Atg8 with Atg19 is mutually exclusive, which may confer directionality to the system.

## Introduction

Macro-autophagy (hereafter autophagy) is a conserved pathway for the delivery of cytoplasmic material into the lysosomal system for degradation (*Kraft and Martens, 2012*). Upon induction of autophagy, double membrane organelles termed autophagosomes are formed in a de novo manner. Initially, autophagosome precursors appear as small membrane structures referred to as isolation membranes (or phagophores). As isolation membranes expand, they gradually enclose cytoplasmic cargo material. Upon closure of the isolation membranes, autophagosomes are formed within which the cargo is isolated from the rest of the cytoplasm. Autophagosomes subsequently fuse with the lysosomal compartment where the inner membrane and the cargo are eventually degraded.

When induced by starvation autophagy can be relatively non-selective with regard to the cargo that is sequestered within autophagosomes. However, it has become clear that autophagy can be highly selective and even exclusive when induced by the presence of intracellular cargo material (*Zaffagnini and Martens, 2016*). Substances including aggregated proteins, cytosolic pathogens and damaged or surplus organelles have all been shown to be selectively degraded by autophagy (*Khaminets et al., 2016*). Autophagy thereby protects the organism from pathological conditions such as neurodegeneration, cancer and infection (*Levine and Kroemer, 2008*; *Mizushima and Komatsu, 2011*). In *S. cerevisiae* the cytoplasm-to-vacuole-targeting (Cvt) pathway mediates the delivery of the oligomeric prApe1 enzyme as well as Ams1 and Ape4 into the vacuole via small auto-phagosomes that are referred to as Cvt vesicles (*Nakatogawa et al., 2009*).

Selectivity of autophagic processes is mediated by cargo receptors that link the cargo to isolation membranes due to their ability to simultaneously bind the cargo and Atg8-family proteins on the

**eLife digest** A living cell must remove unhealthy or excess material from its interior in order to remain healthy and operational. Cells pack this waste into membrane-bound compartments named autophagosomes in a process called autophagy. So-called autophagy proteins make sure that only the unwanted material is eliminated. However, it was not completely clear how these proteins achieve this.

What was known was that proteins called cargo receptors recognize and bind to specific waste materials. At the same time, so-called autophagy enzymes tag the membranes of the autophagosome with a protein known as Atg8, so that cargo receptor molecules can bind this membrane. Now, Fracchiolla, Sawa-Makarska et al. report that, in yeast, an autophagy enzyme links these two events by binding to the cargo receptor and promoting the tagging of the autophagosome's membrane at the same place. The enzyme in question is a complex made from three autophagy proteins (called Atg12, Atg5 and Atg16), and its activity ensures that the membrane is tagged only next to those materials that need to be disposed of.

Although it is now clearer how a cell's waste ends up in the autophagosome, it is still puzzling how this process is regulated and how the other autophagy-related components contribute to this highly coordinated process. In particular, an important next step will be to find out what is the source of membrane that gives rise to the autophagosome.

isolation membrane (*Johansen and Lamark, 2011*; *Rogov et al., 2014*; *Stolz et al., 2014*). The interaction of the cargo receptors with Atg8-family proteins is mediated by LC3-interacting regions (LIRs) (*Pankiv et al., 2007;Ichimura et al., 2008*) also known as Atg8 interacting motifs (AIMs) in the cargo receptors (*Noda et al., 2010*). Atg8-family proteins are ubiquitin-like proteins that are conjugated to the headgroup of the membrane lipid phosphatidylethanolamine (PE) rendering the otherwise soluble proteins membrane-bound (*Ichimura et al., 2000*). This conjugation reaction is also referred to as lipidation. The Atg8 conjugation cascade is analogous to the chain of reactions that mediate the conjugation of ubiquitin to its substrates. Thus, Atg8 is activated by the E1-like enzyme Atg7 under consumption of ATP and subsequently transferred to the E2-like enzyme Atg3 from which Atg8 is ultimately transferred to the headgroup of PE (*Ichimura et al., 2000*; *Klionsky and Schulman, 2014*). This last step is strongly facilitated by a complex composed of the Atg12~Atg5 protein conjugate and Atg16. The Atg12~Atg5-Atg16 complex acts in an E3-like manner and determines the site of Atg8 conjugation (*Fujita et al., 2008b*; *Hanada et al., 2007*). The Atg8 conjugation machinery acts in concert with other proteins of the autophagic machinery including the Atg1/ULK1 complex, the class III PI3K complex 1, Atg9 and the WIPIs to mediate the efficient generation of autophagosomes or Cvt vesicles (*Dooley et al., 2014*; *Fujita et al., 2008a*; *Juris et al., 2015*; *Kishi-Itakura et al., 2014*; *Komatsu et al., 2005*; *Kraft et al., 2012*; *Mizushima et al., 1998*, *2001*; *Sou et al., 2008*). The precise mechanisms by which the Atg12~Atg5-Atg16 complex and Atg8 aid the formation, elongation or closure of the autophagosomal membranes are unclear.

Recent work has provided important information about how the presence of an autophagic cargo induces the formation of an isolation membrane. In particular, it was shown that the Atg19 cargo receptor recruits the Atg11 scaffold protein to the prApe1 cargo for Atg1 kinase activation (*Kamber et al., 2015*; *Torggler et al., 2016*). In addition, it was demonstrated that the cargo receptors Optineurin and NDP52 recruit the ULK1 complex to damaged mitochondria (*Lazarou et al., 2015*). Furthermore, TRIM proteins were shown to localize the ULK1, PI3K complexes and ATG16L1 to their cargo in a process referred to as precision autophagy (*Chauhan et al., 2016*; *Kimura et al., 2015*).

A major question is how the presence of an autophagic cargo is coupled to Atg8 conjugation and thus isolation membrane formation in space and time. Here we show that the *S. cerevisiae* Atg19 and Atg34 as well as the human p62, Optineurin and NDP52 cargo receptors interact with the E3-like Atg12~Atg5-Atg16 complex. Employing Atg19 as a model in a fully reconstituted system we show that it is capable of recruiting Atg12~Atg5-Atg16 to the prApe1 cargo. This recruitment is

mediated by a direct interaction of the AIM motifs in Atg19 with the Atg5 subunit. In our in vitro system the recruitment of the Atg12~Atg5-Atg16 complex is sufficient to drive accumulation of lipidated Atg8 at the cargo. Since the interaction of the Atg19 cargo receptor with the E3-like Atg12~Atg5-Atg16 complex is outcompeted by Atg8, the system may have an inherent directionality whereby the final product in form of Atg8~PE could displace the upstream conjugation machinery at the concave side of the isolation membrane.

## Results

During classical ubiquitination reactions the localization of the E3 ligase determines where ubiquitin is conjugated to its substrates (*Deshaies and Joazeiro, 2009*; *Komander and Rape, 2012*). We therefore asked if autophagic cargo receptors could interact with the Atg12~Atg5-Atg16 E3-like complex and thereby recruit it to the cargo. Indeed, in pull down experiments GST-Atg19 used as a bait successfully pulled down Atg12~Atg5-Atg16, demonstrating a direct interaction between these two components (*Figure 1A*). In a complementary approach we imaged the recruitment of Atg12~Atg5-Atg16-mCherry to beads coated with GST-Atg19 under equilibrium condition (*Figure 1B*). Atg12~Atg5-Atg16-mCherry was robustly and specifically recruited to these beads (*Figure 1B*). The α-mannosidase (Ams1) receptor Atg34 was also able to bind the Atg12~Atg5-Atg16 complex (*Figure 1C*), suggesting that this interaction is a more general property of cargo receptors. In order to test if this interaction occurs in cells we performed immunoprecipitation experiments using Atg5-TAP to pull down 6xmyc-Atg19 (*Figure 1D*). Atg19 was specifically pulled down by Atg5-TAP. Employing the M-Track assay, which is based on the methylation of the human histone 3 N-terminus by the human SUV39H1 methyltransferase when the two components come into close contact (*Brezovich et al., 2015*; *Zuzuarregui et al., 2012*), we confirmed that Atg19 and the Atg12~Atg5-Atg16 complex are in close proximity in living cells (*Figure 1E*). It was previously shown that overexpression of a methyltransferase does not result in unspecific methylation of the histone 3 N-terminus (*Brezovich et al., 2015*). Next, we tested if the interaction of cargo receptors with the E3-like complex is conserved. To this end, we co-expressed human GFP-ATG5 and mCherry-p62 in HeLa cells in which the endogenous p62 had been knocked-down by RNAi. mCherry-p62 was efficiently co-precipitated by GFP-ATG5 (*Figure 1F*). We confirmed this result by using a microscopy-based assay in which we imaged the recruitment of mCherry-p62 to GFP-ATG5 coated beads in HeLa cell lysates (*Figure 1G*). We extended this analysis by investigating other human cargo receptors and found that NDP52 was also pulled down by GFP-ATG5 (*Figure 1H*). In addition, we detected a weak but consistent co-precipitation of Optineurin (OPTN) (*Figure 1H*). In summary, the *S. cerevisiae* Atg19 and Atg34 cargo receptors directly interact with the Atg12~Atg5-Atg16 E3-like enzyme and an interaction with this complex is also detectable for the human cargo receptors p62, OPTN and NDP52.

Further focusing on Atg19, we tested which of the Atg12~Atg5-Atg16 complex subunits interacts with the Atg19 cargo receptor by using GST-Atg19 as bait to pull down Atg5 stabilized with the N-terminal helix of Atg16 (Atg5-Atg16 (1–46)), the Atg12~Atg5 conjugate, the Atg5-Atg16 complex and the full Atg12~Atg5-Atg16 complex (*Figure 2A* and *Figure 2—figure supplement 2A*). Atg12 could not be tested in isolation since we were unable to purify the protein. All proteins tested showed interaction with Atg19 suggesting that Atg5 is sufficient for the binding to Atg19 (*Figure 2A,B*). We confirmed this result in size exclusion chromatography experiments using Atg5-Atg16 (1–46) and Atg19. Indeed, a fraction of Atg5-Atg16 (1–46) shifted to higher molecular weight fractions in the presence of Atg19 (*Figure 2—figure supplement 1*).

The presence of Atg16 had a stimulatory effect on the interaction of Atg5 with Atg19 suggesting that Atg16 could directly interact with Atg19 (*Figure 2A,B* and *Figure 2—figure supplement 2B*). However, when tested in isolation Atg16 did not show any detectable binding to Atg19 (*Figure 2C*). Full length Atg16 may therefore enhance the interaction by an allosteric effect on Atg5 or by increasing the avidity of the interaction due to its ability to self-associate (*Fujioka et al., 2010*; *Kaufmann et al., 2014*).

In order to identify the regions in Atg19 that are required for the interaction with Atg12~Atg5-Atg16 we tested a series of Atg19 truncation mutants (*Figure 2D*) for their interaction with the entire complex and components thereof (*Figure 2E*). Atg5 and Atg5-Atg16 showed robust interaction with all Atg19 truncations including the C-terminal domain encompassing amino acids 365–415

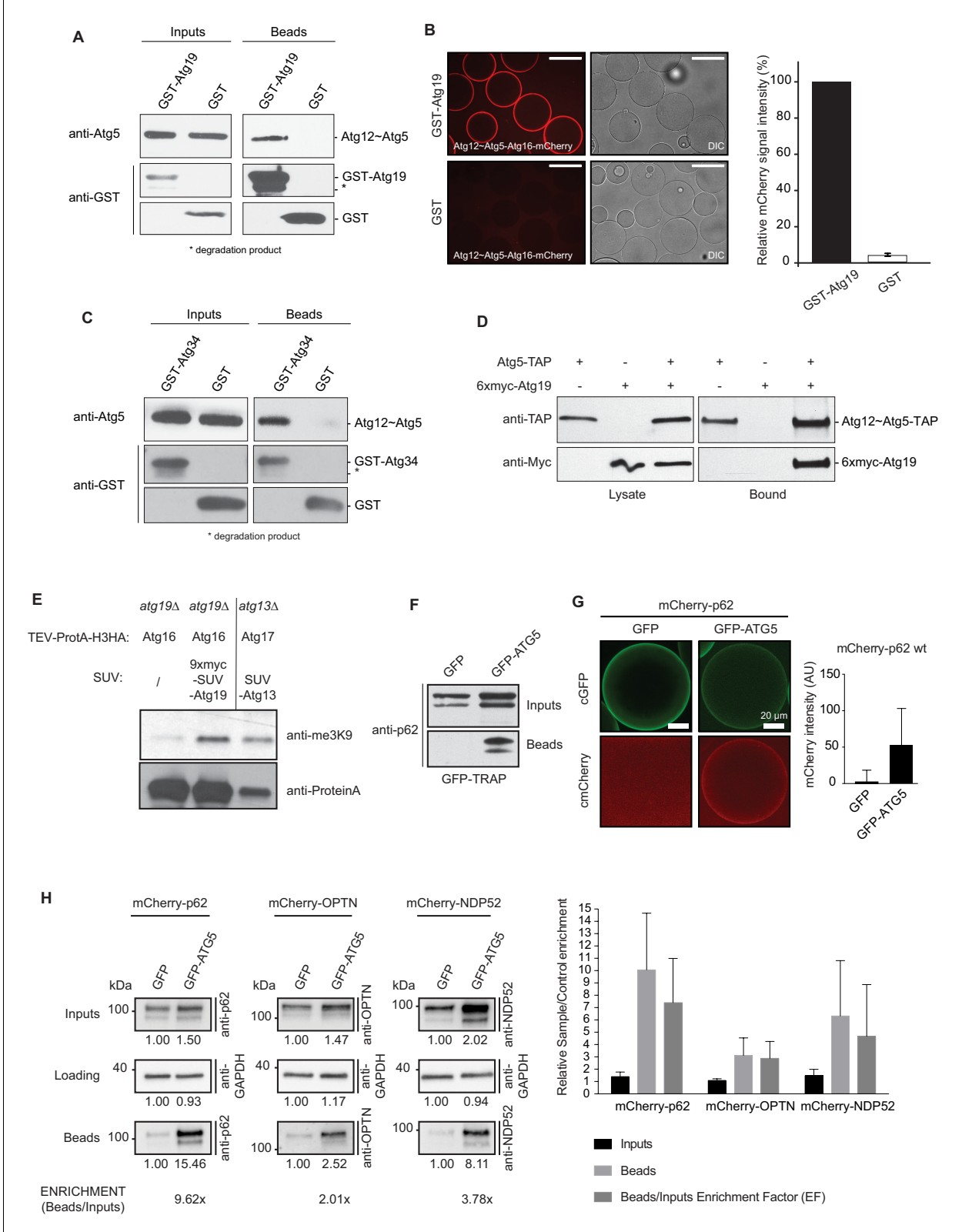

**Figure 1.** Cargo receptors interact with the Atg12~Atg5-Atg16 complex in vitro and in vivo. (**A**) Western blots of GST-pull down experiments using GST-Atg19 as bait and the Atg12~Atg5-Atg16 complex as prey. Degradation bands of GST-Atg19 are marked with an asterisk (*). (**B**) Glutathione Sepharose beads were coated with GST-Atg19 or GST and imaged in the presence of the Atg12~Atg5-Atg16-mCherry complex at equilibrium. The quantification shows the relative mCherry signal intensity measured at the bead in percent. Three independent experiments were considered for

*Figure 1 continued on next page*

*Figure 1 continued*

quantification. Scale bar: 100 µm. (**C**) Same assay as shown in (**A**) but using GST-Atg34 as a bait. (**D**) Western blots of a co-immunoprecipitation experiment using *atg19Δ, atg8Δ S. cerevisiae* cells with integrated Atg5-TAP and transformed with 6xmyc-Atg19. Atg5-TAP was precipitated using magnetic Epoxy IgG-beads. (**E**) M-Track assay using Protein A-Histone 3 (H3)-tagged Atg16 and Atg19 fused to 9xmyc and the SUV39H1 methyl-transferase (HKMT). Shown is a Western blot with an anti-trimethylation-specific antibody to assess the methylation signal and anti-ProteinA to assess the amount of cleaved protein A-H3 on beads. The Atg13 interaction with Atg17 was used as a positive control for the assay. (**F**) Co-immunoprecipitation experiment using GFP-TRAP beads incubated with lysates from HeLa cells transfected with the indicated expression constructs. Endogenous p62 was down regulated by RNAi. (**G**) Lysates from HeLa cells transfected with GFP and mCherry-p62 or GFP-ATG5 and mCherry-p62 were incubated with GFP-TRAP beads and the recruitment of the proteins to the beads was imaged by spinning disc microscopy. The graph shows the average and standard deviation over all beads from one experiment. The endogenous p62 was downregulated by RNAi. (**H**) Western blot analysis of lysates from HeLa cells co-transfected with the indicated constructs and subjected to anti-GFP immunoprecipitation (GFP-TRAP, Chromotek). Numbers below each blot indicate the relative band intensity for the particular blots shown. The beads/input enrichment factors (EF) indicate the fold of enrichment of each mCherry-tagged cargo receptor in the GFP-ATG5 beads fraction over its correspondent GFP control, normalized on the input levels and equalized to the GAPDH blots. Representative blots of at least four independent experiments are shown (left). The plot shows the average sample/control fold enrichment in the indicated fractions for each cargo receptor. The beads/input enrichment factor is defined as above. Averages and standard deviations of at least four independent experiments are shown (right).

The following figure supplement is available for figure 1:

**Figure supplement 1.** Human ATG5 pulls down p62 from cell lysates.

(*Figure 2E* and *Figure 2—figure supplement 2C*). The presence of Atg12, either in context of the Atg12~Atg5 conjugate or the Atg12~Atg5-Atg16 complex, changed the properties of the interaction and required the presence of amino acids 124–254, which include the cargo binding domain of Atg19 (*Yamasaki et al., 2016*). We corroborated the results of the pull down experiments for the full Atg12~Atg5-Atg16 complex in a microscopy-based assay under equilibrium conditions (*Figure 2F*). This assay confirmed that the coiled-coil domain of Atg19 is required for the interaction when Atg5 is conjugated to Atg12 (*Figure 2F*). To interrogate the role of the C-terminal region of Atg19 we performed further microscopy-based interaction experiments (*Figure 2G*). Consistent with the pull down experiments the Atg12~Atg5-Atg16 complex bound strongly to full length Atg19 but not to the isolated C-terminus (amino acids 365–415) (compare *Figure 2E and G*). Intriguingly, deletion of the last eight amino acids containing the canonical AIM motif (*Noda et al., 2008*; *Sawa-Makarska et al., 2014*) from the C-terminus of Atg19 strongly reduced the interaction (*Figure 3A*) suggesting that this AIM motif contributes to the interaction of Atg19 with the Atg12~Atg5-Atg16 complex. Next, we further dissected the contribution of the AIM motifs in the Atg19 C-terminus to the interaction with Atg5. While deletion of the canonical AIM motif in the extreme C-terminus resulted in strong but incomplete reduction in Atg5 binding, further mutation of two additional AIM-like sequences (*Abert et al., 2016*; *Sawa-Makarska et al., 2014*) completely abolished the interaction (*Figure 3B* and *Figure 3C Figure 3—figure supplement 1*). The AIM-dependent association of Atg19 with Atg5 is relevant for the interaction of the two proteins in vivo as the W412A mutation in the canonical AIM motif of Atg19 results in markedly reduced interaction of the two proteins in co-immunoprecipitation experiments (*Figure 3D*).

The dependence of the Atg19 - Atg12~Atg5-Atg16 interaction on the AIM motifs suggested that it is mutually exclusive with the interaction of Atg19 with Atg8. To test this possibility, we immobilized the C-terminus of Atg19 on beads and added Atg5-mCherry-Atg16 (1–46). Subsequently, we added GFP-Atg8 or Atg8 to the beads and determined the signal of Atg5-mCherry-Atg16 (1–46) on the beads (*Figure 3E*). Indeed, Atg8 outcompeted the Atg19-bound Atg5-mCherry-Atg16 (1–46) in a dose dependent manner. The loss of the Atg5-mCherry signal correlated with an increased GFP-Atg8 signal at the bead (*Figure 3E,F*). Thus, the interaction of Atg5 with the C-terminus of Atg19 is AIM-dependent and mutually exclusive with Atg8 binding. Next, we asked if the interaction of Atg19 with Atg5 is also mutually exclusive with the Atg19 - Atg8 interaction in the context of the entire Atg12~Atg5-Atg16 complex and when Atg19 is bound to the prApe1 cargo. First, we generated cargo mimetic beads by attaching the prApe1 propeptide to them via a GST-tag. Consistent with previous results Atg19 was robustly recruited to these beads (*Figure 3—figure supplement 2A*) (*Pfaffenwimmer et al., 2014*; *Sawa-Makarska et al., 2014*). Next we tested if Atg19 recruited the Atg12~Atg5-Atg16 complex to the artificial cargo. Indeed, the Atg12~Atg5-Atg16 complex

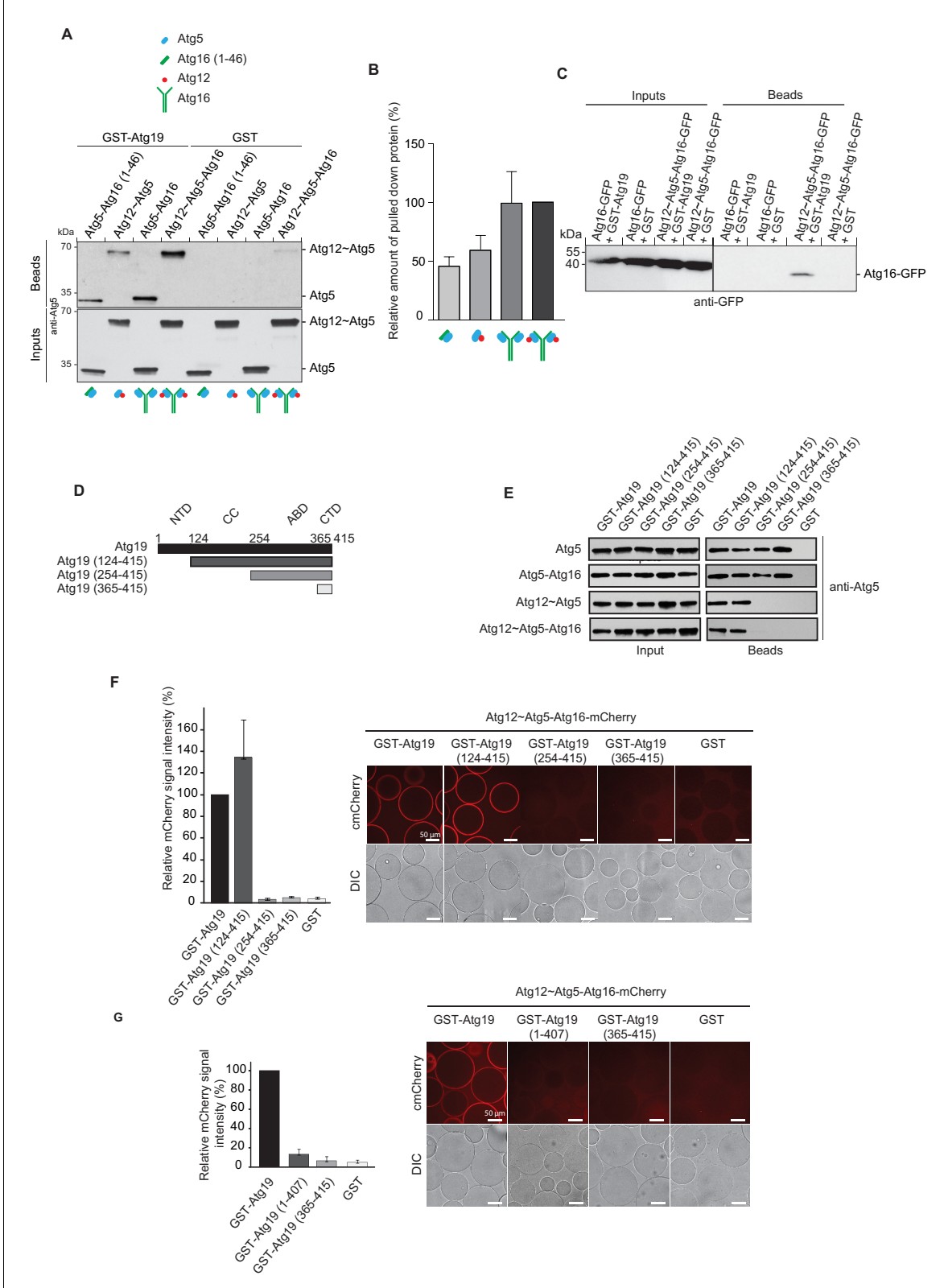

**Figure 2.** Atg19 directly binds Atg5 via its C-terminal domain and requires its coiled-coil domain to interact with the Atg12~Atg5-Atg16 complex. (**A**) GST-pull down experiment using GST-Atg19 or GST as bait in the presence of recombinant Atg5-Atg16 (1–46), Atg12~Atg5, Atg5-Atg16 or Atg12~Atg5-Atg16 complexes as preys. Input and bead fractions were loaded on a SDS-PAGE gel and subjected to Western blotting. Proteins were detected using an anti-Atg5 antibody. See also *Figure 2—figure supplements 1* and *2*. (**B**) Quantification of GST-pull down experiments, one of which

*Figure 2 continued on next page*

*Figure 2 continued*

is shown in (**A**). The amount of pulled down protein for the Atg12~Atg5-Atg16 complex was set to 100%. Average values were calculated from three independent experiments and plotted in the histogram together with the standard deviations. (**C**) GST-pull down experiment using GST-Atg19 or GST as bait and recombinant Atg16-meGFP or the Atg12~Atg5-Atg16-meGFP complex as prey. Input and bead samples were loaded on a SDS-PAGE gel and subjected to Western blotting. Proteins were detected using an anti-GFP antibody. See also *Figure 2—figure supplement 2*. (**D**) Schematic representation of the Atg19 domain organization. N-terminal domain (NTD, residues 1–124), coiled-coil domain (CC, residues 124–254), Ams1 binding domain (ABD, residue 254–365) and C-terminal domain (CTD, residues 365–415). (**E**) The Atg12~Atg5-Atg16 complex and its subunits, Atg5-Atg16 (1–46), Atg5-Atg16 and the Atg12~Atg5 conjugate were incubated with either full length GST-Atg19 or truncated versions thereof as baits and the pulled down protein was detected by Western blotting using an anti-Atg5 antibody. The bead fractions showing GST-labeled Atg19 and truncated versions thereof are depicted in *Figure 2—figure supplement 2*. (**F**) Glutathione beads coated with full length GST-Atg19 or truncations thereof were incubated with the Atg12~Atg5-Atg16-mCherry complex and its recruitment to the beads was determined by spinning disc microscopy. A quantification of three independent experiments is shown to the left. The signal measured for binding to GST-Atg19 full length was set to 100%. (**G**) Glutathione beads were coated with Atg19 full length (GST-Atg19), Atg19 C-terminal domain (GST-Atg19(365–415)) or Atg19 lacking the C-terminal canonical AIM motif (GST-Atg19(1–407)) and incubated with Atg12~Atg5-Atg16-mCherry. A quantification of three independent experiments is shown to the left. The signal measured for binding to GST-Atg19 full length was set to 100%.

The following figure supplements are available for figure 2:

**Figure supplement 1.** Atg19 and Atg5 interact in solution.

**Figure supplement 2.** Mapping Atg19 interaction with Atg12~Atg5-Atg16 complex.

showed a strong Atg19 dependent signal at the beads (*Figure 3—figure supplement 2B*). Employing this experimental setup, we then went on by adding GFP-Atg8 to the cargo mimetic beads (*Figure 3G*). The complex was displaced in a concentration dependent manner confirming that Atg12~Atg5-Atg16 and Atg8 compete for the same binding sites on Atg19.

The Atg5 subunit of the Atg12~Atg5-Atg16 complex and the AIM-like motifs in Atg19 are both essential for the binding of these two components. Moreover, the interaction of Atg19 with Atg5 and Atg8 is mutually exclusive, strongly suggesting that the AIM motifs in Atg19 directly interact with Atg5. To identify potential binding sites in Atg5 for the AIM motif we employed computational modeling and molecular dynamics simulations. To this end, we used the structure of Atg5-Atg16 (1–46) from *S. cerevisiae* (PDB:2DYO) (*Matsushita et al., 2007*) and performed molecular dynamics simulations which allowed us to capture the flexibility of the Atg5-Atg16 (1–46) complex. Subsequently, in silico docking was performed for 10 randomly selected snapshots from the molecular dynamics trajectory using a peptide encompassing the canonical AIM motif ([411]TWEEL[415]) of Atg19. Modelling analysis suggested three possible binding sites for the peptide, two of which mapped to Atg5 (*Figure 4A,B*) and one to a site formed by Atg16. Since our pull down experiments (*Figure 2C*) did not detect any direct interaction of Atg16 with Atg19 the latter site was excluded from further analysis. The other two sites were analyzed. Both contained residues that were persistently involved in forming salt bridges and/or hydrogen bonds with the TWEEL peptide. These were K57 and K137 in the first binding site (*Figure 4A*, pose 1) and N84 and R208 in the second binding site (*Figure 4A*, pose 2). These two sites show a similar architecture composed of a hydrophobic pocket surrounded by positively charged residues. Specifically, molecular dynamics simulations showed that the hydrophobic pocket serves to dock W412 and L415 of the TWEEL peptide, while lysine, arginine and asparagine residues on the surface of Atg5 contact the peptide via salt bridges and hydrogen bonds with the glutamic acid residues E413 and E414. Structural superposition showed that the two sites are unrelated to the AIM-binding sites of Atg8 (*Figure 4—figure supplement 1*).

In order to validate the predictions from the molecular dynamics simulations we set out to interfere with the interaction. To this end, we mutated the predicted binding sites in Atg5 at residues K57, K137, N84 or R208 to E. We refrained from mutating the hydrophobic pockets of Atg5 in order to avoid non-specific loss of function effects due to disruption of the hydrophobic core of the protein. The mutant proteins were tested for their ability to bind to full length GST-Atg19 in pull down experiments (*Figure 4C*). Atg5 K137E as well as Atg5 R208E still efficiently bound to Atg19, while the Atg5 K57E and Atg5 N84E mutants showed no detectable binding in this assay.

We went on to test the effect of the K57E and N84E mutations on the recruitment of the Atg12~Atg5-Atg16 complex to cargo mimetic beads (*Figure 5A–C*). The wild type Atg12~Atg5-

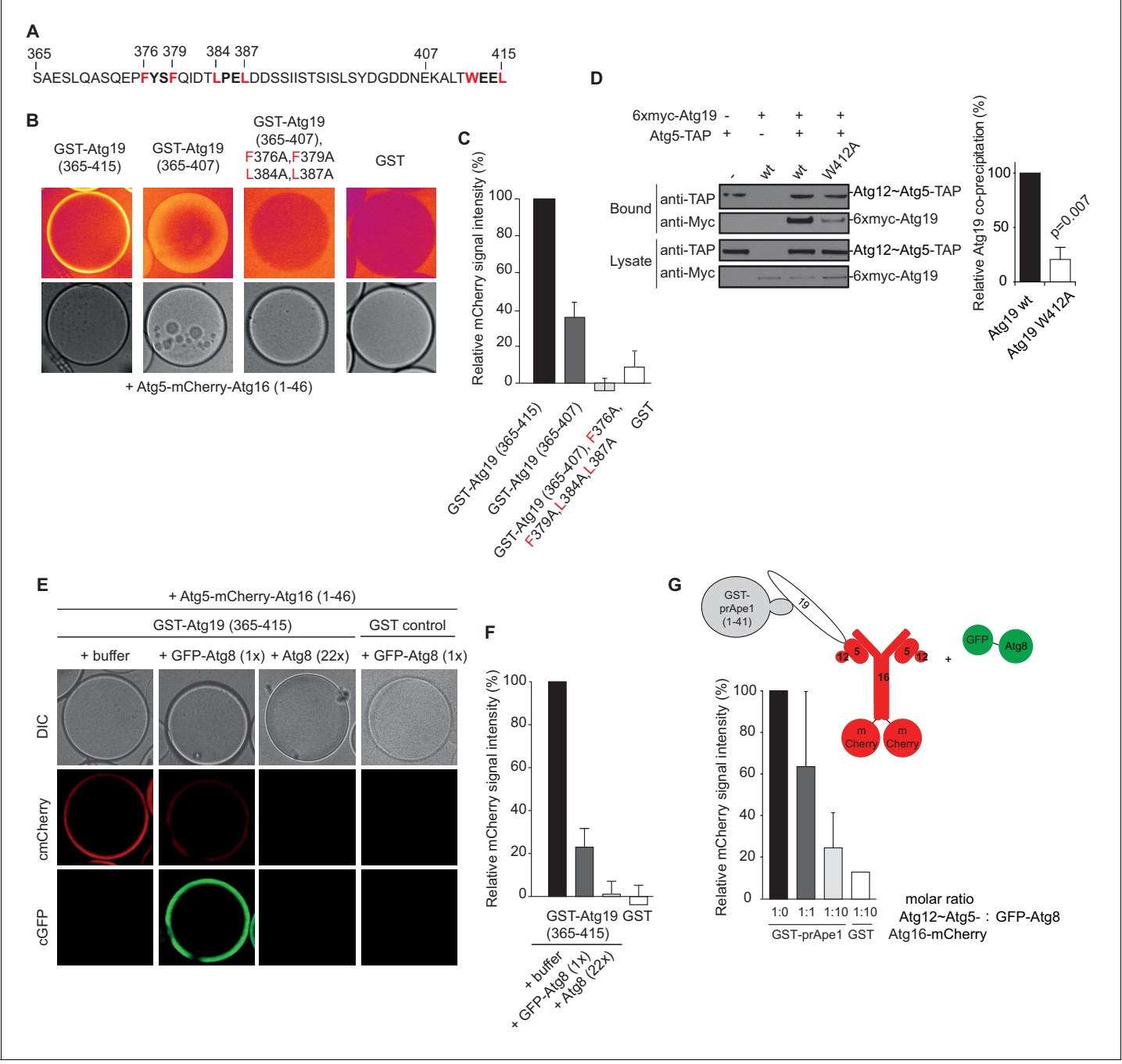

**Figure 3.** The AIM motifs in the C-terminal domain of Atg19 are required for its interaction with Atg5 and are competitively bound by Atg8. (**A**) Amino acid sequence of the C-terminal domain (365–415) of Atg19 containing the canonical AIM motif ([412]WEEL[415]) and two AIM-like motifs ([376]FYSF[379], [384]LPEL[387]). (**B**) Glutathione beads coated with the indicated proteins were imaged in the presence of Atg5-mCherry-Atg16 (1–46) complex under equilibrium conditions. The mCherry signal is shown in false color (ImageJ: Fire). See also *Figure 3—figure supplement 1*. (**C**) Quantification of three independent experiments of the relative mCherry signal intensity measured at the bead. Due to optical reasons very low signals at the beads resulted in values lower than the background and thus negative values. Error bars represent standard deviations. (**D**) Co-immunoprecipitation experiment of 6xmyc-Atg19 or 6xmyc-Atg19W412A with Atg5-TAP as bait in *S. cerevisiae* cell lysates. Western blots of bead and lysate fractions are shown. Atg12~Atg5-TAP was detected with an anti-TAP and 6xmyc-Atg19 with an anti-Myc antibody. A quantification of three independent experiments is shown to the right. Shown are averages and standard deviations. (**E**) Glutathione beads coated with the GST-Atg19 C-terminus (365–415) or GST were imaged in the presence of Atg5-mCherry-Atg16 (1–46) complex under equilibrium conditions. For the competition experiment recombinant GFP-Atg8 (or buffer) was added to the sample at a final concentration corresponding to 1x initial concentration of Atg5-mCherry-Atg16 (1–46). Purified Atg8 (or buffer) was added to a final concentration of 22x the initial concentration of Atg5-mCherry-Atg16 (1–46)). Representative microscopy pictures are

*Figure 3 continued on next page*

*Figure 3 continued*

shown. (**F**) Quantification of three independent experiments, one of which is shown in (**E**). The mCherry intensity in the '+ buffer' sample were GST-Atg19(365–415) was used as bait was set to 100%. Due to optical reasons very low signals at the beads resulted in values lower than the background and thus negative values. Bars represent standard deviations. (**G**) Glutathione Sepharose beads were coated with GST-prApe1(1–41) and Atg19 or GST and imaged in the presence of the Atg12~Atg5-Atg16-mCherry complex. Subsequently, recombinant GFP-Atg8 was added to the sample at a final concentration corresponding to 1x or 10x the initial concentration of the Atg12~Atg5-Atg16-mCherry complex. The binding of the complex to the beads in the absence of Atg8 was set to 100%. The histogram shows the averaged values of three independent experiments and the error bars represent standard deviations. N = 3 for the prApe1 samples. See also *Figure 3—figure supplement 2*.

The following figure supplements are available for figure 3:

**Figure supplement 1.** AIM-dependent interaction of the Atg19 C-terminal domain with Atg5.

**Figure supplement 2.** Recruitment of Atg19 and Atg12~Atg5-Atg16 to cargo mimetic beads.

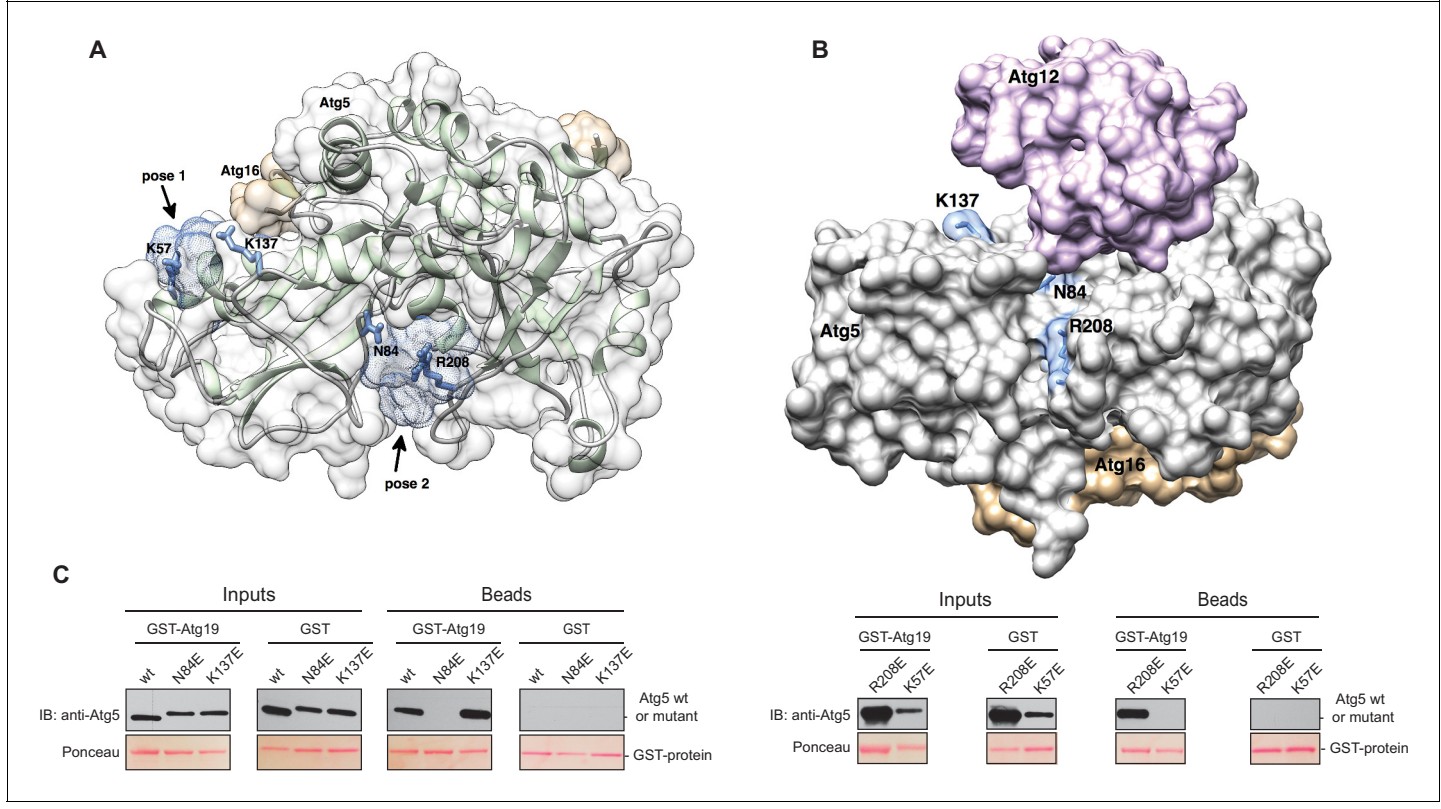

**Figure 4.** Structure of the Atg5-Atg16 (1–46) complex with the predicted binding sites for the TWEEL peptide. (**A**) Structure of Atg5 in complex with the N-terminus of Atg16 (PDB:2DYO, [*Matsushita et al., 2007*]). The backbone of Atg5 is shown in green and Atg16 is shown in orange. The position of the predicted TWEEL pentapeptide binding sites on Atg5 are shown in blue (pose 1 and 2). The residues predicted to contact the glutamates of the TWEEL peptide are shown as sticks. (**B**) Surface representation of the Atg12~Atg5-Atg16 (1–46) complex (PDB:3W1S [*Noda et al., 2013*]). The amino acids in the potential TWEEL binding sites predicted to contact the glutamates of the peptide are shown in blue. Atg5 is shown in grey, Atg12 is shown in violet and Atg16 is shown in orange. (**C**) Representative blots of GST-pull down experiments conducted with GST-Atg19 or GST in the presence of Atg5-Atg16 (1–46) wild type and its single mutant versions (K57E, N84E, K137E and R208E) are shown. Input and bead samples were subjected to Western blot analysis. GST-fusion bait proteins were detected with Ponceau Staining (lower row) and prey proteins were detected with an anti-Atg5 antibody (upper row).

The following figure supplement is available for figure 4:

**Figure supplement 1.** Structural relationship between the Atg5 and Atg8 AIM binding sites.

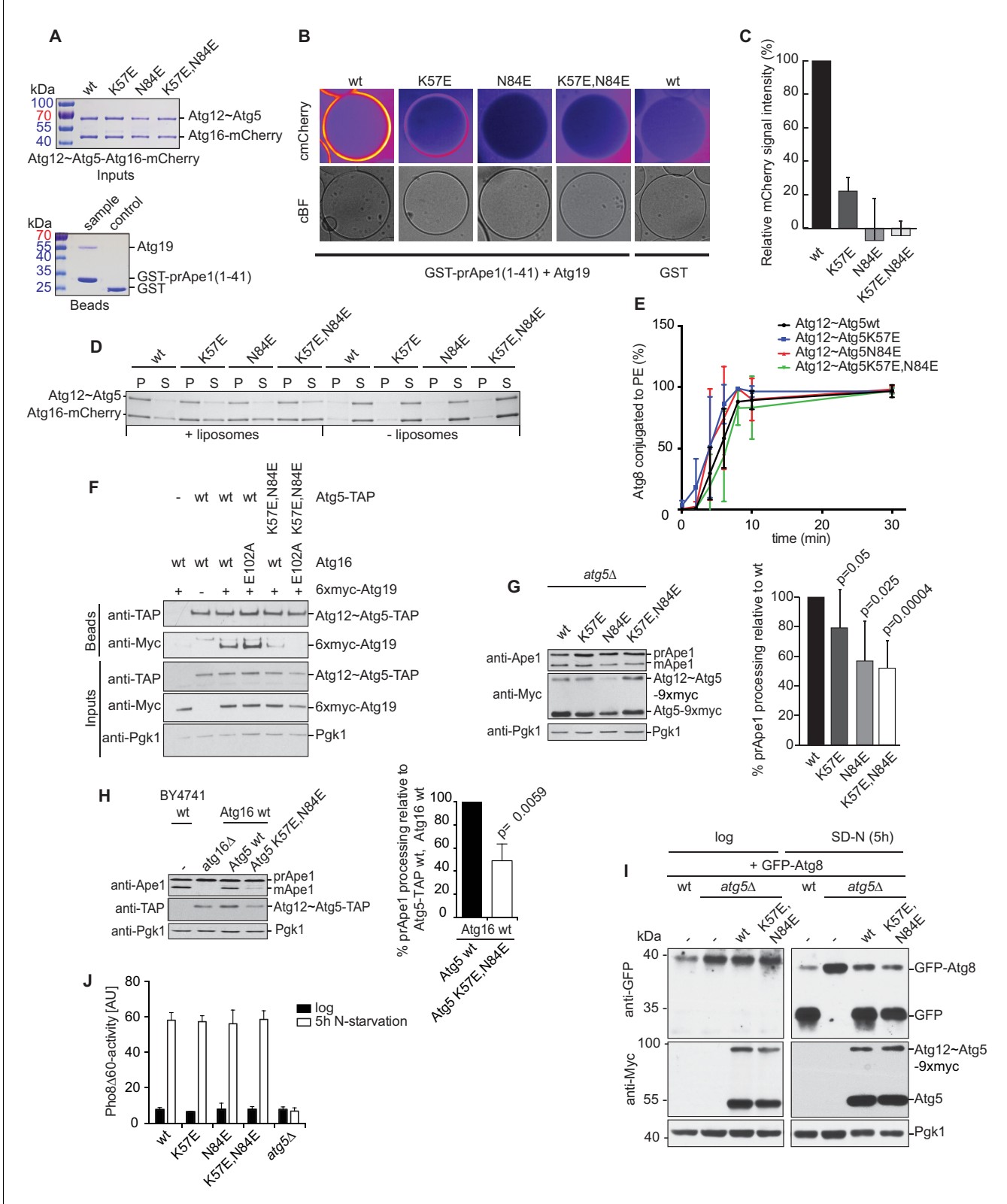

**Figure 5.** Mutation of the predicted binding sites for Atg19 in Atg5 impairs the recruitment of the Atg12~Atg5-Atg16 complex to cargo mimetic beads and Cvt pathway function but does not affect bulk autophagy. (**A**) Coomassie stained gels showing the input amounts of the Atg12~Atg5-Atg16-mCherry complex (upper gel) and of the GST-prApe1(1–41) + Atg19 or GST proteins on the beads (lower gel) used for the experiment shown in (**B**). (**B**) GST-prApe1(1–41) + Atg19 or GST coated glutathione beads imaged in the presence of Atg12~Atg5-Atg16-mCherry complex (wild type, K57E, N84E

*Figure 5 continued on next page*

*Figure 5 continued*

or K57E,N84E). (**C**) Quantification of three independent experiments of the relative mCherry signal intensity measured at the beads. One experiment used for the quantification is shown in (**B**). The signal measured for the wild type Atg5~Atg12-Atg16-mCherry complex was set to 100%. Due to optical reasons very low signals at the beads resulted in values lower than the background and thus negative values. Error bars represent standard deviations. (**D**) Coomassie-stained gel showing the result of a liposome co-sedimentation assay using wild type Atg12~Atg5-Atg16-mCherry and the indicated point mutants thereof. Liposome binding allows the protein to be pelleted (P). The unbound protein remains in the supernatant (S). (**E**) Quantification of in vitro Atg8 conjugation assays using the indicated mutants of Atg12~Atg5. The amount of conjugated Atg8 and un-conjugated Atg8 was measured as the band intensity signal on a Coomassie stained gel and set as 100%. Amounts of conjugated Atg8 were determined relative to this. Averages of these values were calculated from three independent experiments and the final values are plotted together with standard deviations. See also *Figure 5— figure supplement 1*. (**F**) Co-immunoprecipitation using Atg5-TAP or Atg5 K57E,N84E-TAP as bait in the presence of 6xmyc-Atg19 and either Atg16 wild type or Atg16 E102A. 6xmyc-Atg19 pulled down protein in wild type Atg5 and Atg16 expressing samples was set to 100. All the other conditions were quantified in relation to this (Atg5 wild type and Atg16 E102A, 85% (±59) p-value = 0.63, n.s.; Atg5 K57E,N84E and Atg16 wild type, 13% (±12.9) p-value < 0.0001; Atg5 K57E,N84E and Atg16 E102A, 2.9% (±4.6) p-value < 0.0001). The p-values were calculated using a two-tailed Student t-test. Proteins were detected using anti-TAP and anti-Myc antibodies, anti-Pgk1 was used as loading control. Shown is a representative blot of three experiments. (**G**) prApe1 processing assay using an *atg5Δ* strain transformed with the indicated expression constructs. The lower Ape1 band indicates prApe1 processing and thus its delivery into the vacuole. The prApe1 and Ape1 bands were detected with an anti-Ape1 antibody. The expression levels of Atg5 were visualised with an anti-Myc antibody. The Pgk1 signal served as a loading control. The bar graph to the right shows a quantification of six independent experiments. The p-values were calculated using a two-tailed Student t-test. (**H**) prApe1 processing assay using yeast *atg16Δ* strain with Atg5 wild type-TAP or Atg5 K57E,N84E-TAP stably integrated in the genome and transformed with the indicated Atg16 constructs. The blot shows the prApe1 processing in the Atg5 mutants in combination with Atg16 wild type in rich conditions (the full set of tested Atg16 can be found in *Figure 5—figure supplement 2*). A blot showing the full set of Atg16 mutants after 6 hr nitrogen-starvation is shown in *Figure 5—figure supplement 3*. The Ape1 bands were detected using an anti-Ape1 antibody. The bar graph shows quantification of the prApe1 processing of four independent experiments. The p-value was calculated using a two-tailed Student t-test. (**I**) A representative blot of a GFP-Atg8 cleavage assay is shown. (**J**) Pho8Δ60-activity assay under rich (black bars) and 5 hr N-starvation (white bars) growing conditions using a *pho13Δ, pho8Δ60, atg5Δ* strain transformed with the indicated Atg5 expression constructs or an empty vector. At least three independent experiments were conducted and the mean values for each conditions were plotted. The error bars represent the standard deviation.

The following figure supplements are available for figure 5:

**Figure supplement 1.** The K57E and N84E mutations do not impair the ability of the Atg12~Atg5 conjugate to promote Atg8 conjugation.

**Figure supplement 2.** Effect of the disruption of the Atg16 – Atg21 interaction on prApe1 processing.

**Figure supplement 3.** The Cvt pathway is affected upon disrupting Atg19 – Atg5 interaction.

**Figure supplement 4.** Mutation of the predicted binding sites for Atg19 in Atg5 does not affect bulk autophagy.

Atg16-mCherry complex robustly localized to the beads. Introduction of the K57E into Atg5 resulted in a strongly reduced recruitment of the Atg12~Atg5-Atg16 complex, while the N84E mutation rendered the recruitment undetectable (*Figure 5B,C*). When combined, the two mutations also resulted in a loss of Atg12~Atg5-Atg16-mCherry complex recruitment to the beads. Thus, introducing negative charge at positions 57 and 84 in the two predicted AIM binding sites interfered with the interaction of the two proteins. These residues are therefore likely to be directly involved in the formation of binding sites for the AIM motif. Interestingly, in the context of the Atg12~Atg5 conjugate N84 in pose two would be largely covered by Atg12 (*Figure 4B*). This may explain why the C-terminal domain of Atg19 containing the AIM motif is not sufficient for the interaction with Atg12~Atg5 and requires the coiled-coil domain (*Figure 2D–G*), which may reorient Atg12 away from pose 2.

The K57 and N84 mutations did not abolish the conjugation of Atg5 to Atg12 (*Figure 5A*). We also did not observe significant defects of the mutant Atg12~Atg5-Atg16 complexes with respect to liposome binding (*Figure 5D*) or of the single and double mutant Atg12~Atg5 conjugates with respect to promoting Atg8 lipidation (*Figure 5E* and *Figure 5—figure supplement 1*).

We went on to test if the K57E,N84E double mutation would interfere with the Atg5 - Atg19 interaction in vivo by performing co-immunoprecipitations (*Figure 5F*). Consistent with the results shown above (*Figure 3D*), wild type Atg5 robustly co-precipitated Atg19. The K57E,N84E double mutant Atg5 showed a strongly reduced ability to interact with Atg19, but the interaction was still detectable (*Figure 5F*). It was previously shown that Atg16 interacts with Atg21 and that this interaction recruits the Atg12~Atg5-Atg16 complex to the pre-autophagosomal structure (PAS)

(*Juris et al., 2015*). We therefore asked if the residual interaction of the K57E,N84E mutant Atg5 with Atg19 could be dependent on the recruitment of the Atg12~Atg5-Atg16 complex to the PAS by Atg21. To this end, we introduced the E102A mutation into Atg16, which was reported to abolish the Atg16 - Atg21 interaction (*Juris et al., 2015*). Indeed, the interaction of the Atg5 K57E,N84E with Atg19 became undetectable in context of the Atg16 E102A mutation (*Figure 5F*).

Next, we tested the impact of the single mutations or their combination on the functionality of the Cvt pathway by monitoring prApe1 processing. The N84E mutation, which had the stronger effect on the Atg19 - Atg5 interaction also affected prApe1 processing more pronouncedly compared to the K57E mutation while the K57E,N84E double mutation had the strongest effect on prApe1 processing (*Figure 5G,H* and *Figure 5—figure supplement 2*). For the cells expressing Atg5-TAP (*Figure 5H* and *Figure 5—figure supplement 2*) we consistently noticed a somewhat lower levels of the Atg12~Atg5 for the K57E,N84E double mutant under rich conditions. This effect was not seen for the myc-tagged version. prApe1 processing was also affected by the K57E,N84E double mutation under starvation condition in the context of the Atg16 D101A, E102A mutation (*Figure 5—figure supplement 3 Figure 5H*). In contrast, starvation-induced bulk autophagy as measured by GFP-Atg8 cleavage and the Pho8Δ60 assay was not significantly affected by the Atg5 K57E,N84E double mutation (*Figure 5I,J*), even when tested in combination with the Atg16 D101A, E102A mutant (*Figure 5—figure supplement 4*).

The data presented so far have shown that the Atg19 cargo receptor can recruit the E3-like Atg12~Atg5-Atg16 complex to the prApe1 cargo and that mutations that abolish this interaction reduce the efficiency of the Cvt pathway. The cargo receptor - E3 interaction might therefore be a minimal axis to couple Atg8 conjugation to the cargo. To test this hypothesis, we developed a fully reconstituted system to recapitulate these reactions. In analogy to the experiment shown in *Figure 3—figure supplement 2*, we added the Atg12~Atg5-Atg16 complex to cargo mimetic beads bound by Atg19. The Atg12~Atg5-Atg16 complex directly binds to membranes (*Figure 5D*) (*Romanov et al., 2012*) and it may thus be able to link membranes and the cargo. To test this, we added small unilamellar vesicles (SUVs) labeled by the incorporation of ATTO390-PE or Rhodamine to Atg19-bound cargo mimetic beads in the presence or absence of the Atg12~Atg5-Atg16 complex (*Figure 6A*, *Figure 6—figure supplement 1*). The SUVs were recruited to the beads in an Atg12~Atg5-Atg16 complex dependent manner (*Figure 6A*, *Figure 6—figure supplement 1*). The complexes containing the Atg19 binding defective mutants of Atg5 were less efficiently recruited to cargo mimetic beads and were also less efficient in membrane recruitment (*Figure 6—figure supplement 2*). Thus, the complex brings the membrane substrate for Atg8 conjugation in proximity of the cargo. Next we analyzed if this minimal system would allow local accumulation of conjugated Atg8 by adding the ubiquitin-like molecule GFP-Atg8, the E1-like enzyme Atg7, the E2-like enzyme Atg3, and the cofactors $MgCl_2$ and ATP to the system. Intriguingly, GFP-Atg8 showed increased signal at the cargo in the presence of ATP (*Figure 6A*). We also detected an increased signal for the membrane and the Atg12~Atg5-Atg16 complex (*Figure 6A*), likely due to its association with Atg8 and the membranes (*Kaufmann et al., 2014*; *Romanov et al., 2012*). Since the presence of ATP increased the GFP-Atg8 signal at the beads, this effect may be due to lipidation of Atg8 and thus its stable anchoring and concentration on the membranes. If so, then the lipidated Atg8 should be more strongly bound by the Atg19 receptor (*Abert et al., 2016*; *Sawa-Makarska et al., 2014*). To test this, we conducted FRAP experiments on the cargo mimetic beads (*Figure 6B*). Indeed, Atg8 recovered more slowly in the presence of ATP and recovery was slowest for the Atg8 positive puncta, which we interpret as larger Atg8-positive vesicular structures (*Figure 6B*).

In the experiments shown in *Figure 6A,B* the vesicles were also present in solution and we could therefore not exclude the possibility that conjugation occurred remotely from the cargo, followed by attachment of the vesicles harboring lipidated Atg8. For this reason, we recruited vesicles to cargo mimetic beads via the Atg12~Atg5-Atg16 complex and washed away unbound vesicles (*Figure 6C*). Subsequently, we added the Atg8 conjugation machinery. In the presence of ATP the bead associated signal was increased, consistent with local lipidation of Atg8 at the cargo (*Figure 6C*). To corroborate this result, we performed FRAP experiments on these beads (*Figure 6D*). Indeed, the recovery of Atg8 was much reduced in the presence of ATP, consistent with its stable attachment to the membrane via lipidation. Furthermore, upon addition of the Atg4 protein, which removes Atg8 from PE the GFP-Atg8 signal decreased (*Figure 6—figure supplement 3*).

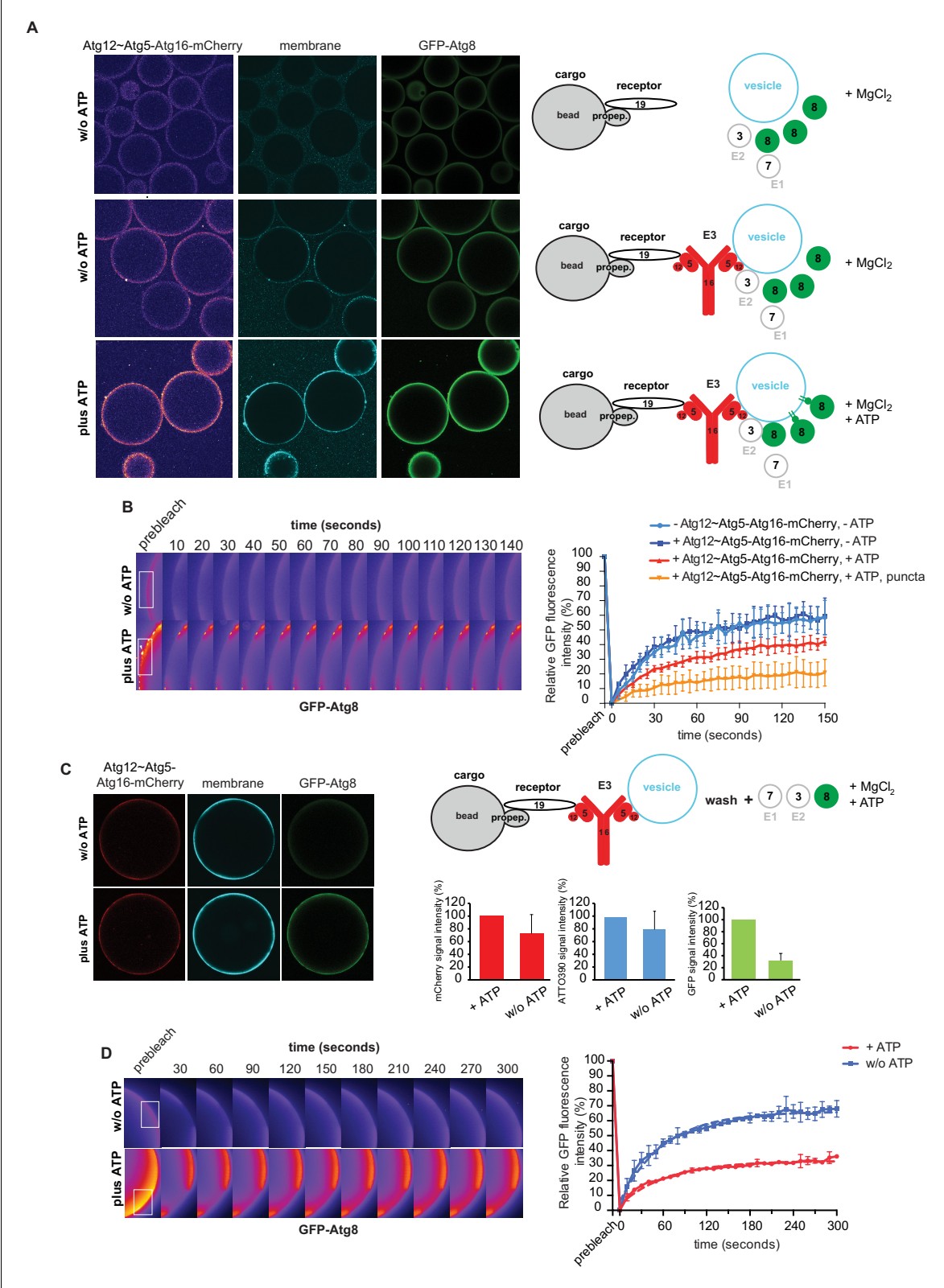

**Figure 6.** In vitro reconstitution of cargo-directed Atg8 lipidation. (**A**) GST-prApe1(1–45) + Atg19 ± Atg12~Atg5-Atg16-mCherry coated Sepharose beads were imaged in the presence of ATTO390-containing SUVs and GFP-Atg8ΔR117, Atg7, Atg3, MgCl$_2$ with or without ATP. Representative pictures and the experimental scheme are shown. The images corresponding to the mCherry channel are displayed in false color (ImageJ: Fire). See also *Figure 6—figure supplement 1* and *Figure 6—figure supplement 2*. (**B**) Representative pictures of a GFP-Atg8 FRAP experiment conducted on the

*Figure 6 continued on next page*

*Figure 6 continued*

surface of GST-prApe1(1–45) + Atg19 + Atg12~Atg5-Atg16-mCherry coated Sepharose beads after the conjugation reaction performed in the presence (lower row) or absence (upper row) of ATP. The graph shows the quantification of the GFP signal measured on the surface of at least two beads per condition (Cx: Atg12~Atg5-Atg16-mCherry). (C) Representative pictures, experimental scheme and quantification of Atg8 lipidation on cargo mimetic beads coated with GST-prApe1(1–45) + Atg19 + Atg12~Atg5-Atg16-mCherry + ATTO390-containing SUVs in the presence of the conjugation machinery (Atg7, Atg3, GFP-Atg8ΔR117, MgCl$_2$, ±ATP) after removal of SUVs from the solution by washing. (D) Representative pictures of a GFP-Atg8 FRAP experiment conducted on the surface of GST-prApe1(1–45) + Atg19 + Atg12~Atg5-Atg16-mCherry + ATTO390-containing SUVs-coated Sepharose beads, after conjugation reaction performed in the presence (lower row) or absence (upper row) of ATP. The graph shows the quantification of the recovered GFP signal on the beads in the presence or absence of ATP, respectively, for at least two beads per condition. See also *Figure 6—figure supplement 3*.

The following figure supplements are available for figure 6:

**Figure supplement 1.** Atg19 AIM-dependent recruitment of the Atg12~Atg5-Atg16 complex and vesicles to cargo mimetic beads.

**Figure supplement 2.** Vesicle recruitment by Atg12~Atg5-Atg16 mutants to cargo mimetic beads.

**Figure supplement 3.** De-conjugation of lipidated Atg8 on the beads.

## Discussion

Here we have shown that the cargo receptor Atg19 directly interacts with the Atg5 subunit of the E3-like Atg12~Atg5-Atg16 complex and that this interaction is sufficient to direct Atg8 conjugation to the cargo in a reconstituted system. The recruitment of the Atg12~Atg5-Atg16 complex requires the AIM motifs of Atg19 and it is likely that these motifs directly bind Atg5 since our modeling analysis and molecular dynamics simulations predicted two binding sites for the C-terminal AIM motif of Atg19 and mutation of these sites abolished Atg19 binding. In addition, the interaction of Atg19 with Atg5 is competitive with the interaction with Atg8, which also directly interacts with the C-terminal AIM motif of Atg19 (*Noda et al., 2008*). Additional regulation of the two mutually exclusive binding events may occur due to phosphorylation as it was shown that Atg19 is phosphorylated in its C-terminal domain (*Pfaffenwimmer et al., 2014*; *Tanaka et al., 2014*).

Since Atg19 contains multiple AIM-like sequences, it is possible that both sites in Atg5 contribute to Atg19 binding. Mutation of N84 in site two had a more pronounced effect on the interaction with Atg19. In the context of the Atg12~Atg5 conjugate this site would be buried by the Atg12 subunit (*Noda et al., 2013*). Since the coiled-coil domain of Atg19 was required for the interaction of Atg19 with Atg5 when conjugated to Atg12, we hypothesize that the coiled-coil domain is required to expose binding site two by changing the position of Atg12.

The interaction of cargo receptors with the Atg12~Atg5-Atg16 complex is conserved since we detected an interaction of the *S. cerevisiae* Atg34 and human p62, NDP52 and Optineurin cargo receptors with Atg12~Atg5-Atg16 and ATG5, respectively, suggesting that the recruitment of the E3-like ligase for ATG8-family members conjugation to the cargo is a more general property of cargo receptors. Future work will have to elucidate the biochemical details of the interaction of the human cargo receptors with ATG5 and the ATG12~ATG5-ATG16L complex. It is possible that this interaction is at least for p62 in part mediated indirectly by ALFY since it interacts with p62 and ATG5 (*Filimonenko et al., 2010*). A similar mechanism has been described for *C. elegans* where EPG-7 binds both, p62/SQST-1 and ATG12/LGG-3 (*Lin et al., 2013*).

The results presented in this study suggest the following sequence of events, at least for the Cvt pathway (*Figure 7*). When bound to their respective cargo, cargo receptors cluster and provide a high-avidity binding platform that recruits the autophagy machinery, including the E3-like Atg12~Atg5-Atg16 complex. This machinery is able to bring and keep membranes in close proximity to the cargo and to act catalytically by promoting several rounds of Atg8 conjugation. Consistent with the idea of local Atg8 lipidation the E2-like Atg3 enzyme localizes to the site of autophagosome formation (*Ngu et al., 2015*). The Atg12~Atg5-Atg16 complex is able to directly bind lipids (*Romanov et al., 2012*) and thus it may link the cargo to the membrane. However, in vivo additional interactions with PROPPINs/WIPIs will render the system more robust (*Dooley et al., 2014*; *Juris et al., 2015*).

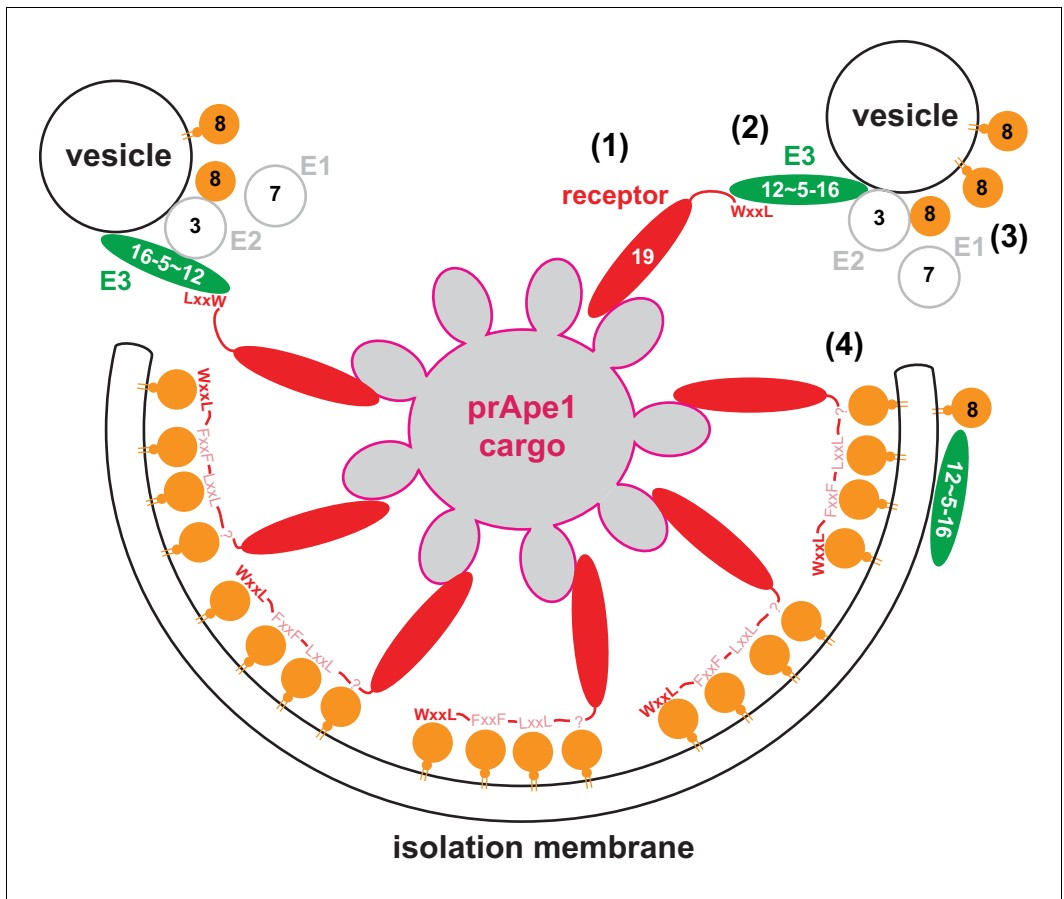

**Figure 7.** Model for the molecular mechanism of cargo-directed Atg8 lipidation. The Atg19 cargo receptor binds the cargo (1) and recruits the E3-like Atg12~Atg5-Atg16 complex to the cargo via its AIM motifs (2). In the presence of a membrane source, Atg8 is locally conjugated (3) and may eventually outcompete the Atg12~Atg5-Atg16 complex from Atg19 binding. At the same time the AIM motifs of Atg19 may keep the Atg8-coated isolation membrane close to the cargo excluding non-cargo material from incorporation into selective autophagosomes (4). See main text for extended discussion.

Due to the action of the Atg8 conjugation machinery Atg8 accumulates at the isolation membrane and may subsequently outcompete the Atg12~Atg5-Atg16 complex on the concave side of the isolation membrane. In conflict with this hypothesis is the finding that the signal of the Atg12~Atg5-Atg16 complex at the bead was not decreased upon addition of ATP (*Figure 6C*). We interpret this result with the previously described interactions of the Atg12~Atg5-Atg16 complex with the membrane and lipidated Atg8 on the convex side (*Kaufmann et al., 2014*; *Romanov et al., 2012*). Consistent with exclusion of Atg12~Atg5-Atg16 from the concave side of the isolation membrane, the Atg12~Atg5-Atg16 complex is excluded from the autophagosomal lumen in vivo (*Mizushima et al., 2003*). On the concave side Atg8 may be subsequently bound with high avidity by the cargo receptors, which could result in close apposition of the membrane and the cargo and thus exclusion of non-cargo material from the autophagosome (*Figure 7*) (*Abert et al., 2016*; *Sawa-Makarska et al., 2014*; *Wurzer et al., 2015*). As a consequence, this mechanism would localize the Atg8 conjugation machinery to the highly curved edge of the membrane where the isolation membrane has not yet formed, which may additionally stimulate Atg8 conjugation (*Nath et al., 2014*). Thus, the intricate AIM/LIR-based interplay between the cargo, the cargo receptors, the Atg8 conjugation machinery and Atg8 may serve to confer directionality to the Atg8 lipidation resulting in robust membrane growth exclusively around the cargo. In vivo, this may occur at a permissive site such as the vacuole or the endoplasmic reticulum (*Lamb et al., 2013*; *Nakatogawa et al., 2009*).

## Materials and methods

### Accession numbers

Atg3: NP_014404; Atg4: NP_014176.2; Atg5: NP_015176.1; Atg7: NP_012041.1; Atg8 NP_009475.1; Atg10: NP_013058.1; Atg12: NP_009776.1; Atg16: NP_013882.1; Atg19: NP_014559.1; Atg34: NP_014558.1; prApe1: NP_012819; p62/SQSTM1: NP_003891; NDP52: AAA75297.1; OPTN: NP_001008212.

### Protein expression and purification

A list of constructs for protein expression can be found in (*Table 1.*). Expression and purification of Atg19 and shortened variants thereof, Atg34 and prApe1(1–45) was described previously (*Sawa-Makarska et al., 2014*). prApe1(1–41) was subcloned into pGEX4T-1 vector, expressed and purified as a GST fusion protein with the same approach as the propeptide prApe1(1–45) variant described in (*Sawa-Makarska et al., 2014*). The mCherry-Atg19 was purified via a hexahistidine tag as described in (*Sawa-Makarska et al., 2014*).

Atg12~Atg5-Atg16, Atg12~Atg5, Atg16 and Atg16-meGFP as well as all proteins required for Atg8~PE conjugation (Atg3, Atg7, Atg10, Atg8ΔR117, meGFP-Atg8ΔR117) were expressed and purified as described previously (*Romanov et al., 2012*).

Atg5-Atg16 (1–46) for analytical size exclusion chromatography was produced by co-expressing Atg5 subcloned into pGEX4T-3 and the first 46 amino acids of Atg16 (Atg16 (1–46)) subcloned into pCOLADuet-1 in *E. coli* Rosetta pLySS. The co-transformed cells were grown at 37°C to an OD600 of 0.6, induced with 0.1 mM IPTG and further grown over night at 18°C. Cells were pelleted and resuspended in a buffer containing 50 mM HEPES pH 7.5, 300 mM NaCl, 1 mM $MgCl_2$, 2 mM β−mercaptoethanol, complete protease inhibitors (Roche, Basel, Switzerland) and DNAse I (Sigma, USA, Missouri). Cells were disrupted by freeze-thawing and lysates were cleared by ultracentrifugation (140,000 *g* for 30 min at 4°C in a Ti45 rotor Beckman, Brea, CA, USA). Supernatants were applied to glutathione beads (GE Healthcare, Buckinghamshire, UK) for 1 hr at 4°C. Beads were washed five times with 50 mM HEPES, 300 mM NaCl, 1 mM DTT and the protein was cleaved from the GST tag by incubation with thrombin protease (SERVA, Heidelberg, Germany) overnight at 4°C. The supernatant containing Atg5 - Atg16 (1–46) was concentrated and applied to a Superdex 200 (16/60 prep grade, GE Healthcare) and eluted with a buffer containing 25 mM HEPES at pH 7.5, 150 mM NaCl and 1 mM DTT. Fractions containing pure protein were pooled, concentrated, frozen in liquid nitrogen and stored at −80°C.

For all other experiments, wild type Atg5 as well as Atg5(K57E), Atg5(N84E), Atg5(K137), Atg5 (R208) and Atg5(K57E,N84E) were tagged with an N-terminal hexahistidine-tag followed by a TEV cleavage site. The proteins were co-expressed and purified in complex with Atg16 (1–46) as described in (*Romanov et al., 2012*).

Atg16-mCherry with an N-terminal hexahistidine-tag followed by a TEV cleavage site (pETDuet-1) was expressed in *E. coli* Rosetta pLysS. Cells were grown at 37°C to an OD600 of 0.6 and induced with 0.1 mM IPTG. The protein expression continued overnight at 16°C. Cells were pelleted and resuspended in a buffer containing 50 mM HEPES pH 7.5, 300 mM NaCl, 10 mM Imidazole, 1 mM $MgCl_2$, 2.5 mM β−mercaptoethanol, complete protease inhibitors (Roche) and DNAse I (Sigma). Cells were lysed by freeze-thawing followed by 30 s sonication and the lysate was centrifuged at 40000 rpm (Beckman Ti45 rotor) for 40 min at 4°C. The supernatant was applied to a 5 ml Ni-NTA column (GE Healthcare) and eluted via a stepwise imidazole gradient (50, 75, 100, 150, 200, and 300 mM). Protein-containing fractions were pooled and subjected to overnight cleavage with TEV protease at 4°C in the dark. The cleaved protein was applied to a Superdex 200 column (16/60 prep grade, GE Healthcare, Sweden) and eluted with a buffer containing 25 mM HEPES pH 7.5, 500 mM NaCl and 1 mM dithiothreitol (DTT). Fractions containing the purified proteins were pooled, concentrated, frozen in liquid nitrogen, and stored at −80°C.

For purification of Atg5-Atg16 and Atg5-Atg16-meGFP, Atg5 was expressed as a GST-tagged protein in *E. coli* Rosetta pLysS from pGEX4T-3 vector. The expression and purification followed the same procedure as for Atg5-Atg16 (1–46). In short, cells were grown at 37°C to an OD600 of 0.5 and induced with 100 μM IPTG for 16 hr at 18°C. Cells were disrupted by freeze-thawing and the cleared lysate was incubated with glutathione beads (Glutathione Sepharose 4B, GE

**Table 1.** Table of constructs.

| Identification number | Vector | Expression system | Expressing | Published |
|---|---|---|---|---|
| SMC3 | pGEX-4T-3 | *E. coli* Rosetta pLysS | GST-Atg3 | (*Romanov et al., 2012*) |
| SMC7 | pGEX-4T-3 | *E. coli* Rosetta pLysS | GST-Atg5 | this study |
| SMC17 | pOPTH | *E. coli* Rosetta pLysS | 6xHis-Atg7 | (*Romanov et al., 2012*) |
| SMC34 | pOPTG | *E. coli* Rosetta pLysS | GST-Atg16 | (*Romanov et al., 2012*) |
| SMC58 | pET-Duet-1 | *E. coli* Rosetta pLysS | 6xHis-Atg8ΔR117 | this study |
| SMC126 | pET Duet-1 | *E. coli* Rosetta pLysS | 6xHis-Atg5, Atg12 | (*Romanov et al., 2012*) |
| SMC131 | pCOLA Duet-1 | *E. coli* Rosetta pLysS | Atg7, Atg10 | (*Romanov et al., 2012*) |
| SMC156 | pET Duet-1 | *E. coli* Rosetta pLysS | 6xHis-mCherry-Atg19 | (*Sawa-Makarska et al., 2014*) |
| SMC159 | pGEX-4T-1 | *E. coli* Rosetta pLysS | GST-Atg19 | (*Sawa-Makarska et al., 2014*) |
| SMC178 | pET Duet-1 | *E. coli* Rosetta pLysS | 6xHis-Atg16-meGFP | (*Romanov et al., 2012*) |
| SMC179 | pET Duet-1 | *E. coli* Rosetta pLysS | 6xHis-meGFP-Atg8ΔR117 | (*Romanov et al., 2012*) |
| SMC180 | pCOLA-Duet | *E. coli* Rosetta pLysS | 6xHis-Atg16 (1–46) | (*Romanov et al., 2012*) |
| SMC185 | pGEX-4T-1 | *E. coli* Rosetta pLysS | GST-Atg34 | (*Sawa-Makarska et al., 2014*) |
| SMC188 | pGEX-4T-1 | *E. coli* Rosetta pLysS | GST-Atg19W412A | (*Sawa-Makarska et al., 2014*) |
| SMC293 | pGEX-4T-1 | *E. coli* Rosetta pLysS | GST-Atg19(124–415) | (*Sawa-Makarska et al., 2014*) |
| SMC294 | pGEX-4T-1 | *E. coli* Rosetta pLysS | GST-Atg19(254–415) | (*Sawa-Makarska et al., 2014*) |
| SMC295 | pGEX-4T-1 | *E. coli* Rosetta pLysS | GST-Atg19(365–415) | (*Sawa-Makarska et al., 2014*) |
| SMC300 | pGEX-4T-1 | *E. coli* Rosetta pLysS | GST-prApe1(1–45) | (*Sawa-Makarska et al., 2014*) |
| SMC301 | pGEX-4T-1 | *E. coli* Rosetta pLysS | GST-Atg19(1–407) | (*Sawa-Makarska et al., 2014*) |
| SMC309 | pGEX-4T-1 | *E. coli* Rosetta pLysS | GST-Atg19(365–407) | this study |
| SMC564 | pGEX-4T-3 | *E. coli* Rosetta pLysS | GST-Atg4 | (*Zens et al., 2015*) |
| SMC595 | pGEX-4T-1 | *E. coli* Rosetta pLysS | GST-prApe1(1–41) | this study |
| SMC665 | pET Duet-1 | *E. coli* Rosetta pLysS | 6xHis-Atg5(K57E), Atg12 | this study |
| SMC668 | pET Duet-1 | *E. coli* Rosetta pLysS | 6xHis-Atg5(N84E), Atg12 | this study |
| SMC743 | pET Duet-1 | *E. coli* Rosetta pLysS | 6xHis-Atg5(K57E,N84E), Atg12 | this study |
| SMC772 | pGEX-4T-2 | *E. coli* Rosetta pLysS | GST-Atg19(408–415) | this study |
| SMC782 | pET Duet-1 | *E. coli* Rosetta pLysS | 6xHis-Atg5-mCherry | this study |
| SMC808 | pGEX-4T-1 | *E. coli* Rosetta pLysS | GST-Atg19(365–407) F376A, F379A, L384A, L387A | this study |
| SMC819 | pET Duet-1 | *E. coli* Rosetta pLysS | 6xHis-Atg16-mCherry | (*Romanov et al., 2012*) |
| SMC255 | pEGFP-C1 | HeLa cell line | EGFP-ATG5 | this study |
| SMC398 | pmCherry-C1 | Hela cell line | pmCherry-OPTN | this study |
| SMC516 | pmCherry-C1 | HeLa cell line | mCherry-p62 | (*Wurzer et al., 2015*) |
| SMC539 | pmCherry-C1 | Hela cell line | pmCherry-NDP52 | this study |
| | pRS313 | *S. cerevisiae* | | (*Sikorski and Hieter, 1989*) |
| | pRS315, | *S. cerevisiae* | | (*Sikorski and Hieter, 1989*) |
| | pRS316 | *S. cerevisiae* | | (*Sikorski and Hieter, 1989*) |
| | pRS415 | *S. cerevisiae* | | (*Sikorski and Hieter, 1989*) |
| pAB15 | | *S. cerevisiae* | 9xmycHKMT-ATG19_cyc1term, pRS415, ATG19 Promoter | this study |
| pCK48 (SMC199) | pRS315 | *S. cerevisiae* | GFP-Atg8 | (*Kraft et al., 2008*) |
| pLW38.1 | | *S. cerevisiae* | ATG13-HKMT, YCp111, ATG13 promoter | (*Brezovich et al., 2015*) |
| pLW52 | | *S. cerevisiae* | ATG2-9xmyc-HKMT, pRS415, ATG2 promoter | (*Brezovich et al., 2015*) |
| pTP9 (SMC852) | pRS313 | *S. cerevisiae* | endogenous promoter-6xmyc-Atg19 | this study |
| SMC236 | pRS315 | *S. cerevisiae* | endogenous promotor-Atg16 wild type - terminator | this study |
| SMC270 | pRS316 | *S. cerevisiae* | endogenous promoter-Atg5wild type-9xmyc | (*Romanov et al., 2012*) |

*Table 1 continued on next page*

*Table 1 continued*

| Identification number | Vector | Expression system | Expressing | Published |
|---|---|---|---|---|
| SMC343 | pRS316 | *S. cerevisiae* | 6xmyc-Atg19 wild type | (*Sawa-Makarska et al., 2014*) |
| SMC381 | pRS316 | *S. cerevisiae* | 6xmyc-Atg19W412A | (*Sawa-Makarska et al., 2014*) |
| SMC418 | pRS316 | *S. cerevisiae* | 6xmyc-Atg19 F376A, F379A,W412A | (*Sawa-Makarska et al., 2014*) |
| SMC422 | pRS316 | *S. cerevisiae* | 6xmyc-Atg19 F376A, F379A,P385A,E386A, W412A | (*Sawa-Makarska et al., 2014*) |
| SMC524 | pRS315 | *S. cerevisiae* | 6xmyc-Atg19 wild type | this study |
| SMC678 | pRS315 | *S. cerevisiae* | 6xmyc-Atg19W412A | this study |
| SMC690 | pRS316 | *S. cerevisiae* | endogenous promoter-Atg5K57E-9xmyc | this study |
| SMC691 | pRS316 | *S. cerevisiae* | endogenous promoter-Atg5N84E-9xmyc | this study |
| SMC692 | pRS316 | *S. cerevisiae* | endogenous promoter-Atg5K57E,N84E-9xmyc | this study |
| SMC836 | pRS315 | *S. cerevisiae* | endogenous promotor-Atg16(D101A)-terminator | this study |
| SMC837 | pRS315 | *S. cerevisiae* | endogenous promotor-Atg16(E102A)-terminator | this study |
| SMC838 | pRS315 | *S. cerevisiae* | endogenous promotor-Atg16(D101A,E102A)-terminator | this study |

Healthcare, Uppsala, Sweden). The protein was cleaved off from the beads with thrombin protease (SERVA). Next, the supernatant containing Atg5 was mixed with purified Atg16 or Atg16-meGFP in a molar ratio 1:1 at 4°C for 30 min, concentrated and the resulting Atg5-Atg16 or Atg5-Atg16-meGFP complex was further purified by size exclusion chromatography on Superdex S200 (16/60 prep grade, GE Healthcare).

The wild type and Atg12~Atg5-Atg16 (untagged and -mCherry tagged) as well as the point mutants Atg12~Atg5 (K57E)-Atg16-mCherry, Atg12~Atg5 (N84E)-Atg16-mCherry, Atg12~Atg5 (K57E,N84E)-Atg16-mCherry were purified in two steps. In the first step, Atg12~Atg5 wild type conjugate or point mutants thereof were generated as described in (*Romanov et al., 2012*). Next the conjugates were mixed with purified Atg16 or Atg16-mCherry in a molar ratio 1:1 ratio and incubated on ice for 30 min. The resulting Atg12~Atg5-Atg16 or Atg12~Atg5-Atg16-mCherry complexes were further purified by size exclusion chromatography on Superdex S200 (16/60 prep grade, GE Healthcare).

The Atg5-mCherry was subcloned into pETDuet-1 vector with N-terminal hexahistidine-tag followed by a TEV cleavage site (6xHis-TEV-Atg5-mCherry). The protein was co-expressed as a complex with the first 46 amino acids of Atg16 (Atg16 (1–46)) subcloned into pCOLADuet-1. The *E. coli* Rosetta pLysS cells were co-transformed with 6xHis-TEV-Atg5-mCherry and Atg16(1–46) and grown at 37°C to an OD600 of 0.6, induced with 1 mM IPTG and further grown overnight at 18°C. Cells were pelleted and resuspended in a buffer containing 50 mM HEPES pH 7.5, 300 mM NaCl, 10 mM Imidazole, 1 mM $MgCl_2$, 2 mM β−mercaptoethanol, complete protease inhibitors (Roche) and DNAse I (Sigma). Cells were disrupted by freeze-thawing followed by 30 s sonication. Lysates were cleared by ultracentrifugation (140,000 *g* for 30 min at 4°C in a Beckman Ti45 rotor). Supernatant was applied to a 5 ml Ni-NTA column (GE Healthcare, Sweden) and eluted via a stepwise imidazole gradient (50, 75, 100, 150, 200, and 300 mM). Protein-containing fractions were pooled, concentrated, applied onto a Superdex 200 column (16/60 prep grade, GE Healthcare) and eluted with a buffer containing 25 mM HEPES pH 7.5, 150 mM NaCl and 1 mM dithiothreitol (DTT). Fractions containing the purified proteins were pooled, concentrated, frozen in liquid nitrogen, and stored at −80°C.

Atg4 was expressed and purified as described in (*Zens et al., 2015*).

## Analytical SEC

To probe the direct Atg19 interaction with Atg5-Atg16 (1–46) in solution the analytical size exclusion chromatography was performed. Atg19 and Atg5-Atg16 (1–46) were premixed at 55 µM, concentrated to 580 µM (Amicon Ultra-0.5 ml Centrifugal Filters 3 kDa MWCO, Millipore, Cork, Ireland) and subsequently applied onto a Superose 6 gel filtration column (PC 3.2/30, GE Healthcare) equilibrated with a buffer containing 25 mM HEPES at pH 7.5, 150 mM NaCl and 1 mM DTT. Resulting

fractions were subjected to SDS-PAGE and the protein bands were detected with Coomassie Brilliant Blue staining.

## GST pull down binding assays

To perform GST pull down binding assays GST or GST-fused Atg19 wild type or shortened variants thereof were used as a bait and Atg12~Atg5-Atg16 or Atg12~Atg5-Atg16-meGFP, Atg12~Atg5, Atg5-Atg16 (1–46), Atg5-Atg16 or Atg16-meGFP were used as a prey. The purified GST-fused proteins (5 µM for pull downs in *Figures 1A,C*, *2A and C*; 20 µM for pull downs in *Figure 2E*, *Figure 3—figure supplements 1 Figure 4C,*) and purified GST-free proteins (5 µM) as well as glutathione Sepharose 4B beads (GE Healthcare) were simultaneously incubated for 1 hr at 4°C on a rotating wheel. After washing the beads three times with 25 mM HEPES at pH 7.5, 150 mM NaCl, 1 mM DTT (and 0.1% TritonX100 for pull-downs in *Figures 1A, C*, *2A and C*), the glutathione beads together with bound proteins were subjected to SDS-PAGE. The protein bands were detected either by Coomassie Brilliant Blue staining or Ponceau or by immunoblotting carried out with a mouse monoclonal anti-Atg5 (*Romanov et al., 2012*), mouse anti-GFP (Roche, diluted 1:5000 in 0.5% Milk in TBST, 1% TritonX100) or anti-GST (diluted 1:1000 in 3% Milk in TBST, 1% TritonX100) antiserum used as primary antibodies. Secondary antibodies were used as described in ´prApe1 processing assay´.

## Competition assay

For experiments shown in *Figure 3E*, glutathione Sepharose 4B beads were incubated with a 30 µM GST or GST-Atg19 (365–415) solution for 30 min at 4°C on a rotating wheel and afterwards twice washed in buffer containing 25 mM HEPES at pH 7.5, 150 mM NaCl and 1 mM DTT. 2 µl of these beads were added to a Atg5-mCherry-Atg16 (1–46) solution pre-pipetted in the wells of a 384-wells glass-bottom plate (Greiner Bio One, Frickenhausen, Germany) resulting in a final concentration of 18 µM. After at least 20 min of incubation, samples were imaged as described in the 'Microscopy-based protein-protein interaction assay' section. For the competition experiment GFP-Atg8ΔR117 (or buffer) was added to the wells at a final concentration of 18 µM (1x initial Atg5-mCherry-Atg16 (1–46) concentration) and allowed to compete the Atg5-mCherry-Atg16 (1–46) protein and to bind to the GST-protein for at least 20 min. An equivalent volume of empty buffer was added to the control well in order to account for the dilution factor applied to the sample. After imaging, Atg8 (or buffer) was further added to the same wells at a final concentration of 400 µM (22x initial Atg5-mCherry-Atg16 (1–46) concentration) and allowed to reach the equilibrium of binding for at least 20 min. The samples were then imaged as described in Microscopy-based protein-protein interaction assay´ section.

For experiments shown in *Figure 3G*, glutathione Sepharose 4B beads were incubated with a 10 µM GST or GST-prApe1(1–40) solution for 30 min at 4°C on a rotating wheel and subsequently washed twice in buffer containing 25 mM HEPES at pH 7.5, 150 mM NaCl and 1 mM DTT. The beads were further incubated with a solution of Atg19 at 20 µM for at least 30 min. Beads were washed twice and pipetted directly to a 96-well-plate glass bottom well pre-filled with a solution of Atg12~Atg5-Atg16-mCherry at a final concentration of 5 µM. After imaging, GFP-Atg8 solution was added to the well at a final concentration of 5 µM (ratio 1:1 with initial concentration of Atg12~Atg5-Atg16-mCherry). The reaction was allowed to reach the equilibrium for 20 min and imaged as described above. GFP-Atg8 solution was added to the well at a final concentration of 50 µM (ratio 1:10 with initial concentration of Atg12~Atg5-Atg16-mCherry). The proteins were allowed to reach the equilibrium of binding and imaged immediately.

## Preparation of small unilamellar vesicles (SUVs)

SUVs employed in the Atg8 conjugation assay were composed of 39% POPC (Avanti Polar Lipids (Alabaster, AL, USA), Inc., 850457C, 10 mg/ml), 35% POPS (Avanti Polar Lipids, Inc., 840034C, 10 mg/ml), 21% POPE (Avanti Polar Lipids, Inc., 850757C, 10 mg/ml), 5% PI3P (Avanti Polar Lipids, Inc., 850150P, 1 mg/ml). PI3P stock was prepared by resuspension in $CHCl_3$ and subsequent drying under an argon stream and further drying for 1 hr in a dessicator. PI3P was then resuspended in $CHCl_3$:MeOH:1M HCl (molar ratio 2:1:0.1) and incubated for 15' for protonation. The lipid was again dried under an argon stream and subsequently for one hour in a dessicator, and then resuspended in $CHCl_3$:MeOH (3:1) and dried again under an argon stream. After one wash with

CHCl$_3$, PI3P was resuspended in CHCl$_3$ to a final concentration of 1 mg/ml. Corresponding amounts of the lipid stocks were transferred into a glass vial and mixed well before they were dried under an argon stream. The dried lipids were further dried for an additional hour in a desiccator. Subsequently, the dried lipids were rehydrated with liposome buffer (25 mM HEPES pH 7.5, 137 mM NaCl, 2.7 mM KCl and 1 mM DTT) for 15 min. The lipids were resuspended by tapping and gently sonicated for 2 min in a water bath sonicator. The resuspended SUVs were then extruded 21 times through 0.4 μm membrane followed by extrusion through a 0.1 μm membrane (Whatman, Nucleopore, UK) using the Mini Extruder from Avanti Polar Lipids Inc.. The final SUVs suspension has a concentration of 1 mg lipids/ml buffer. SUVs are stable for 2–3 days when stored at 4°C.

Lipid mixture used for the in vitro reconstitution of Atg8 lipidation on cargo-mimetic beads and for experiment in *Figure 6—figure supplement 2*, was composed of 39–35% DOPC, 35% DOPS, 20% DOPE, 5% PI$_3$P, 1–5% of ATTO390-DOPE and buffer was composed of 25 mM HEPES pH 7.5, 150 NaCl and 1 mM DTT.

Lipid composition of SUVs used in *Figure 6—figure supplement 1* consists of 39% POPC, 35% POPS, 20% POPE, 5% PI$_3$P, 1% of Rhodamine-DOPE and buffer was composed of 25 mM HEPES pH 7.5, 150 NaCl and 1 mM DTT.

## Atg8 conjugation assay using SUVs

The conjugation reactions were performed at 30°C and all buffers, solutions and the SUVs with the exception of the proteins were pre-warmed to this temperature. Atg3 and Atg7 were used at final concentrations of 1 μM, whereas Atg8ΔR117 was used at a final concentration of 5 μM and Atg12~Atg5 wild type and mutants were used at 0.2 μM. ATP was used at a final concentration of 100 μM, while MgCl$_2$ was used at a final concentration of 1 mM. The reactions were stopped by the addition of loading dye (12% SDS, 6% beta-mercaptoethanol, 30% Glycerol, 0.05% Coomassie Brilliant blue G-250, 150 mM Tris-HCl pH 7).

The reactions were run on 11% SDS/polyacrylamide gels containing 4.5 M urea in the separating parts. The gels were then stained with Coomassie staining solution (40% methanol, 10% acetic acid, 0.2% Coomassie Brilliant Blue).

For *Figure 5E*, the gels of three independent experiments were quantified using the Analyze Gel tool of ImageJ. Statistical analysis was done in Prism by multiple t tests (unpaired, two-tailed, Holm-Sidak method). A p-value < 0.05 was considered to be significant.

## Liposome co-sedimentation assay

Small unilamellar vesicles (SUVs) were composed of 35% DOPC, 35% DOPS, 20% DOPE, 5% PI$_3$P, 5% of ATTO390-DOPE (Avanti Polar Lipids, Inc.) and prepared as described above. After the drying step the lipids were resuspended in 150 mM NaCl, 50 mM HEPES pH 7.5, 1 mM DTT buffer. For the Atg12~Atg5-Atg16-mCherry and point mutants thereof binding to lipids, 25 μl of freshly prepared SUVs were mixed with 5 μg of protein at the final reaction volume of 50 μl in 150 mM NaCl, 50 mM HEPES pH 7.5, 1 mM DTT buffer. The reaction was incubated for 30 min at room temperature. Next, the liposome bound protein was pelleted by ultracentrifugation for 10 min at 100,000xg at 22°C. Supernatants and pellets were separated and equal amounts were applied on 12% SDS/polyacrylamide gel and visualized by Coomassie Brilliant Blue staining (40% ethanol, 10% acetic acid, 0.2% Coomassie Brilliant Blue).

## In vitro reconstitution of Atg8 lipidation on cargo-mimetic beads

For *Figure 6A,B*: Glutathione Sepharose 4B beads were coated with GST-prApe1(1–45) at a final concentration of 25 μM, incubated for 30 min at 4°C and washed twice with buffer containing 25 mM HEPES at pH 7.5, 150 mM NaCl and 1 mM DTT. Beads were further incubated with Atg19 at a final concentration of 15 μM, washed two times and further incubated with Atg12~Atg5-Atg16-mCherry at a final concentration of 12.5 μM. After 2x washings, 2 μl of the beads were pipetted into the well of a 384-wells plate, pre-filled with 22 μl buffer and subsequently 1 μl ATTO390-containing SUVs was added to the reaction. Conjugation reaction in *Figure 6A,B* was conducted in the presence of 0.5 mM MgCl$_2$, 0.3 μM Atg7, 0.3 μM Atg3 and 0.1 μM meGFP-Atg8ΔR117 with/without 1 mM ATP over night at 4°C with gentle mixing on an orbital shaker.

For the experiments shown in *Figure 6C* and *Figure 6—figure supplement 2*, beads were prepared as for *Figure 6A* regarding the GST-protein, Atg19 and Atg12~Atg5-Atg16-mCherry. Beads were incubated overnight at 4°C under gentle rolling with an excess of ATTO390-SUVs membranes. The day after beads were washed twice using 25 mM HEPES at pH 7.5, 150 mM NaCl and 1 mM DTT buffer (beads pelleted by sedimentation for 10 min on ice) and overnight conjugation reaction was set up as described for *Figure 6A,B*. Imaging was performed using a Zeiss Confocal LSM700 microscope equipped with a 20x/0.8 Plan-Apocromat Objective. The FRAP experiments shown in *Figure 6B and D* were conducted under following conditions: 10 ms FRAP time/pixel; laser beam diameter 10 pixels. Acquisition was performed either every 5 or 10 s.

For the de-conjugation reaction Atg4 was added to the well containing beads, prepared as in *Figure 6A*, at a final concentration of 0.3 µM together with EDTA at a final concentration of 1 mM. Beads were imaged with a Spinning Disk microscope at the indicated time points of reaction.

## Microscopy-based protein-protein interaction assay

For the experiments shown in *Figures 1B*, *2F–G* 20 µl glutathione Sepharose 4B beads slurry (GE Healthcare) were mixed with GST-fused bait proteins (GST-Atg19 or GST-Atg19 variants) to the final concentration of 20 µM and incubated on a rotating wheel at 4°C for at least 30 min. Subsequently the beads were washed twice with 25 mM HEPES at pH 7.5, 150 mM NaCl and 1 mM DTT. In these experiments the prey Atg12~Atg5-Atg16-mCherry was added at the final concentration of 5 µM. After 30 min of incubation at 4°C a 5 µl aliquot of beads was transferred into the well of a 96-well glass-bottom microplate (Greiner Bio-One) pre-filled with 35 µl of 25 mM HEPES at pH 7.5, 150 mM NaCl and 1 mM DTT and immediately imaged with a Spinning Disk microscope.

For the experiments shown in *Figure 3B*, GST-fused proteins were incubated with Sepharose beads at a final concentration of 30 µM and Atg5-mCherry-Atg16 (1–46) was used at a final concentration of 18 µM.

For the experiments shown in *Figure 5B*, Sepharose beads were incubated with GST-proteins at 25 µM, washed twice with 25 mM HEPES at pH 7.5, 150 mM NaCl and 1 mM DTT buffer and further incubated with Atg19 at 15 µM. After two washings, the beads were incubated with Atg12~Atg5-Atg16-mCherry (wild type and mutants) at a final concentration of 8.8 µM.

For the experiments shown in *Figure 3—figure supplement 2* and *Figure 6—figure supplement 1*, GST-prApe1(1–45) was incubated with Sepharose beads at 5 µM concentration. Beads were washed twice and further incubated with Atg19 wild type and mutant at a final concentration of 5 µM. Beads were washed twice and further incubated with Atg12~Atg5-Atg16-meGFP at a final concentration of 5 µM.

The pictures shown in *Figure 1B*, *Figure 2F,G*, *Figure 3B,E*, *Figures 5B* and *6B,D* and those acquired for quantifications (including *Figure 3G* ) were obtained using a LD Achroplan 20x/0.4 Corr Obje*Figure 3*ctive mounted on a confocal spinning disc microscope (Visitron) installed with VisiView 2.1.1 software and processed with ImageJ. To quantify the protein and membrane recruitment to beads the maximum brightness along a straight line drawn through a single bead was taken (maximal fluorescence). Next, the average brightness of an empty portion of each picture was measured (background fluorescence) and subtracted from the maximal fluorescence for each bead. All intensities were normalized to the signal of the wild type protein.

## M-Track assay

The M-Track methylation assay was conducted as previously described (*Zuzuarregui et al., 2012*) (*Papinski et al., 2014*).

## Atg5 - Atg19 co-immunoprecipitation

For blots shown in *Figures 1D* and *3D*, yeast strain BY4741-*atg8Δatg19Δ* with integrated Atg5-TAP was transformed with empty vector pRS315 or vector containing 6xmyc-Atg19 wild type or mutants. Pre-cultures of yeast were grown in selective medium to log phase and then used to inoculate YPD cultures for an overnight growth in log phase. Cells were harvested by centrifugation for 15 min at 3000xg and then washed once with PBS with 2% glucose and 0.5% ammoniumsulfate. Subsequently, the cells were resuspended in a volume of IP-Buffer corresponding to the volume of the pellet (20 mM PIPES pH 6.8, 50 mM KCl, 100 mM K Acetate, 10 mM MgSO$_4$, 10 µM ZnSO$_4$, 1 mM PMSF, 1

mM NaF, 1 mM $Na_3VO_4$, 20 mM beta-GP, 0.5 mM DTT, complete PI tablet (Roche)(two tablets/100 ml solution), 0.1% Triton X100, and frozen in droplets in liquid nitrogen. The cells were then disrupted with a freezer mill (6770; SPEX), the extract was thawed in lysis buffer and cleared by centrifugation. The cleared extract was incubated with 30 µL of magnetic beads (Dynabeads M-270 Epoxy, Invitrogen, Norway) coupled to rabbit IgG from serum (I5006-10MG, Sigma) for 1 hr at 4°C with rotation. The beads were washed five times for 10 min in lysis buffer with rotation and then incubated for 10 min at 95°C with urea loading buffer (116 mM Tris pH 6.8, 4.9% Glycerol, 8 M Urea, 8% SDS).

Proteins were separated by SDS-PAGE and subjected to Western blotting. For detection, rabbit anti-TAP (ThermoScientific, #CAB1001, 1:1 000 for lysates or unbound fractions and 1:10 000 for Co-IP samples respectively in 3% milk/TBST), mouse anti-Myc antibody (clone 4A6, 1:500 in 3% milk/TBST) and mouse-anti Pgk1 (Invitrogen, #459250, California, USA; 1:20 000 in 3% milk/TBST) were used to incubate the Western blots at 4°C overnight. Goat anti-mouse IgG HRP (Dianova, #115–035, Germany; 1:10 000 in 3% milk/TBST) and goat anti-rabbit IgG-HRP (Dianova, #111-035-003, Germany; 1:10 000 in 3% milk/TBST) were used to incubate the membranes for 1 hr at room temperature. All the washing steps were conducted using TBS-Tween 0.1%.

The gels were quantified using the Analyze Gel tool of ImageJ. Values were normalized to the wild type, three independent experiments were quantified. Statistical analysis was done in Prism by Welch's t test. A p-value < 0.05 was considered to be significant.

For blots shown in *Figure 5F*, yeast strain BY4741-*atg16Δatg19Δ* with integrated Atg5-TAP or Atg5 K57E,N84E-TAP was transformed with a vector containing either Atg16 wild type or Atg16 E102A. In addition, they were also transformed with empty vector pRS313 or vector containing 6xmyc-Atg19 wild type. Pre-cultures of yeast were grown in selective medium in log phase and then used to inoculate YPD cultures for a 6 hr growth in log phase. Cells were harvested by filtration on a 90 mm glass filter with pore size 0.45 µm (SterliTech) followed by freezing in liquid nitrogen. Subsequently, a volume of IP-Buffer corresponding to the volume of the pellet was frozen in droplets in liquid nitrogen and added. The cells were then disrupted with a freezer mill (6770; SPEX), and co-immunoprecipitation was performed as described for the Atg5 - Atg19 co-immunoprecipitation with Atg19 mutants above.

## prApe1 processing assay

A list of constructs for the yeast experiments can be found in (*Table 1.*). (*Table 2.*) lists the yeast strains used in this study. For the prApe1 processing assay of Atg5 mutants, yeast strains BY4741 wild type and BY4741-*atg5△* were transformed with empty vector pRS316 or vector containing wild type or mutant Atg5-6xmyc. For the prApe1 processing assay of Atg5 mutants in combination with Atg16 mutants Atg5-TAP and Atg5 K57E,N85E-TAP were integrated stably into the genome of *atg5Δatg16Δ* strains and transformed with empty vector pRS315 or vector containing wild type or mutant forms of Atg16 wild type or mutants.

Pre-cultures of yeast were grown in selective medium to log phase and then used to inoculate complete medium for an overnight growth in log phase. The Atg5 with the Atg16 mutants were additionally subjected to nitrogen-starvation for 6 hr as described in 'GFP-Atg8 cleavage assay'. Whole cell lysates were prepared by trichloroacetic acid extractions. Proteins were separated by SDS-PAGE and subjected to Western blotting. For detection, rabbit-anti Ape1 antiserum (*Romanov et al., 2012*); 1:20000 in 3% milk/TBST), mouse anti-Myc antibody (clone 4A6, 1:1000 in 3%milk/TBST) and mouse-anti Pgk1 (Invitrogen, #459250; 1:20000 in 3%m/TBST) were used to incubate the Western blots at 4°C overnight or at room temperature for 1 hr. Goat anti-mouse IgG HRP (Dianova, #115–035; 1:10000 in 3%m/TBST) and goat anti-rabbit IgG-HRP (Dianova, #111-035-003; 1:10000 in 3%milk in TBST) were used for a subsequent incubation of 30 min at room temperature. All the washing steps were conducted using TBS-Tween 0.1%. Quantification of prApe1 processing was performed using Analyze Gel tool in ImageJ software. Band intensities of pr- and mature Ape1 were measured and the ratio of mature Ape1 to prApe1 was calculated. Values calculated for wild type samples were set to 100%. At least three independent replicates were considered for each set of mutants tested.

## GFP-Atg8 cleavage assay

Yeast wild type and atg5Δ BY4741 strains (*Table 2.*) were co-transformed with empty vector (pRS316) or wild type or mutant K57E,N84E Atg5, together with the GFP-Atg8 expressing plasmid (SMC199). Cells were grown to log-phase in selective medium (Formedium, UK) and further grown for an overnight in log phase in the same selective medium. After two washes in SD-N, cells were transferred to SD-N (Formedium, UK) and subjected to nitrogen starvation for 5 hr. Lysates were prepared as described in ´prApe1 processing assay´ and samples were analysed by Western blotting. Mouse anti-GFP antibody (1:2000 dil. in 3.5% milk/PBST, Roche, Germany) was used for detection of GFP-Atg8. Mouse anti-Myc, anti-Pgk1 antibodies and anti-mouse secondary antibody were used as described in ´prApe1 processing assay´.

## Pho8Δ60 assay

Yeast strains SMy33 and SMy62 (*Table 2.*) were transformed with empty vector pRS316 or pRS316 containing wild type or the mutant Atg5-6xmyc. Pre-cultures were grown to log-phase in selective medium and subsequently transferred to complete medium for an overnight-log-culture. 20 OD-units were taken as Log-aliquots, washed with 0.85% NaCl, 1 mM PMSF and frozen in lysis-buffer (20 mM PIPES pH6.8, 0.5% TritonX100, 50 mM KCl, 100 mM KAcetate, 10 mM MgSO$_4$, 10 µM ZnSO$_4$, 1 mM PMSF, protease inhibitor mix tablet, Roche). The cultures were exposed to nitrogen-starvation for 5 hr and aliquots were washed and frozen as described above. The total proteins were extracted by bead-beating. The concentration of the lysate was determined by Bradford. All lysates were adjusted with lysis buffer to a final concentration of 500 µg/ml.

The enzymatic assay was performed using 4-nitrophenol phosphate powder (Sigma, 71768–5G) diluted to final concentration of 1.25 mM in reaction buffer (0.4% TritonX100, 10 mM MgSO$_4$, 10 µM ZnSO$_4$, 250 mM TrisHCl, pH 8.5) as a substrate. The formation of the product 4-nitrophenol was measured using a spectrophotometer plate-reader at 405 nm. An enzyme blank (composed of lysate and reaction buffer without substrate) was measured and subtracted for every sample. The molarity of the reaction product was determined with a standard curve using 4-nitrophenol 10 mM solution (Sigma, N7660-100ML). An enzyme blank (containing substrate in reaction buffer and lysis buffer without enzyme) was subtracted to the standard curve. The reaction was stopped at a determined time point (same for every sample) between 5 and 25 min with 1M glycine, pH 11 adjusted with 5 M KOH. The activity-units (AU) were calculated using the following formula:

$$activity\ [AU] = \frac{c\ (pNP)\ [nM]}{t[min] * protein\ [mg]}$$

## Cell culture

HeLa cells (CCL-2) were directly purchased from ATCC (Manassas, Virginia, USA) and their identity was not authenticated after purchase. Cells were routinely tested for mycoplasma contamination by PCR (GATC, Konstanz, Germany) and tested negative. Cells were cultured in Dulbecco's modified Eagle medium (DMEM) high glucose, GlutaMAX, pyruvate (Gibco, Waltham, MA, USA) supplemented with 10% heat inactivated fetal bovine serum (FBS, Sigma, MO, USA), 100 units/mL penicillin and 100 µg/mL streptomycin (Gibco) at 37°C and 5% CO$_2$. Cells were used from passage 5 to 20.

## Transient transfection of siRNA and plasmids

A list of constructs for transfections can be found in (*Table 1.*). HeLa cells were seeded to 6-well plates on day 1. Transfection with siRNA against SQSTM1/p62 (sip62) or non-targeting siRNA (siControl) was performed on day 2, transfection with plasmids containing siRNA resistant mCherry-p62 and GFP-ATG5 or GFP was performed on day 4. Cells were lysed on day 5.

For one reaction 50 pmol of sip62 (J-010230–05, Dharmacon, Buckinghamshire, UK) or siControl (D-001810–10, Dharmacon) were pre-incubated with 2.5 µl of Lipofectamine RNAimax (Invitrogen, Waltham, MA, USA) in 500 µl serum-free medium at RT for 20 min. The formed complex was added to cells supplied with 2 ml fresh DMEM containing serum and antibiotics and incubated for two days. Thereafter co-transfection of plasmids containing a siRNA resistant p62 variant in pmCherry-C1 (SMC516, *Wurzer et al., 2015*) and Atg5 in pEGFP-C1 (SMC255) or pEGFP-C1 vector only was performed. 1 to 1.5 µg of plasmid-DNA were mixed with Fugene6 (Promega, WI, USA) in a 1 µg:3 µl ratio (DNA:Fugene6) in serum-free medium and incubated at RT for 15 to 45 min. The mixture was

added to cells supplied with 2 ml fresh DMEM containing serum and Pen/Strep and incubated for 24 hr.

For experiments shown in *Figure 1H* $2 \times 105$ cells/well were seeded in 6-well plates on day 1, and transfected with FuGene6 (Promega) according to manufacturer's instructions on day 2. 0.5 µg of each pmCherry-based vector plus 0.5 µg of empty pEGFP-C1 or 1 µg of pEGFP-ATG5 were employed per well, two wells were transfected per condition. Cells were lysed as described below on day 3.

For lysis cells were washed with cold PBS, 100 µl lysis buffer (20 mM Tris pH 8.0, 10% glycerol, 135 mM NaCl, 0.5% NP-40, complete protease inhibitor (EDTA-free, Roche), 2.5 mM $MgCl_2$, DNase) was added per well and lysis performed for 20 min at 4°C. Lysates from two wells were pooled, cell debris was removed by centrifugation at 16000 g for 10 min at 4°C and supernatant kept frozen at −80°C until use.

Lysates for the microscopy based assay needed to be more concentrated and were therefore prepared by trypsinization of cells followed by a PBS wash and lysis of cell pellets from two pooled wells with 100 µl lysis buffer containing 0.2% NP-40.

## Pull down and microscopy based assay using GFP-TRAP beads

The protein concentration in all HeLa cell lysates was measured with Bradford's method and lysates were normalized to each other accordingly. For the pull-down shown in *Figure 1F*, GFP-Trap_A beads (ChromoTek) were mixed with empty Sepharose beads (Sigma) in a 1:4 ratio and equilibrated in wash buffer (20 mM Tris pH 8.0, 10% glycerol, 135 mM NaCl). Lysates were diluted with 1.5 volumes of wash buffer and incubated with 40 µl equilibrated bead slurry for 1 hr at 4°C with gentle agitation. Beads were washed three times with wash buffer, taken up in 40 µl Laemmli loading buffer, boiled 10 min at 95°C and bound proteins were separated by SDS-PAGE and analysed by Western blotting.

For visualizing protein interaction at equilibrium (*Figure 1G*) 50 µl lysate were incubated with 5 µl equilibrated GFP-Trap_A beads for 1 hr at 4°C. 15 µl of this bead dispersion were added to 20 µl of the corresponding residual lysate prepared in a 96-well plate and imaged using a spinning disc microscope (Visitron).

**Table 2.** Yeast manipulation and strain list.

| Name | Genotype | Background | Source |
|---|---|---|---|
| BY4741/SMy1 | Mat a; his3△1, leu2△0,met15△0, ura3△0 | BY474x | Euroscarf |
| BY4743 | Diploid; his3△1/his3△1, leu2△0/leu2△0, met15△0/met15△0, ura3△0/ura3△0 | BY474x | Euroscarf |
| SMy2 | Mat a; atg5::KanMX | BY474x | Euroscarf |
| SMy33 | △pho8::pho8△60His,△pho13::kan | BY474x | (*Sawa-Makarska et al., 2014*) |
| SMy62 | △Atg5::nat; pho8△60::His; △pho13::KanMX | BY474x | *Sawa-Makarska et al., 2014*) |
| SMy147 | Mat a; atg19::KanMX | BY474x | Euroscarf |
| SMy196 | Mat a; ATG5-TAP:URA3, atg19::KanMX, atg8::NatMX | BY474x | this study |
| SMy201 | Mat a; ATG16-TEV-2xProtA-4xH3-5xHA:URA3; atg19::KanMX | BY474x | this study |
| SMy239 | Mat a; ape1::KanMX | BY474x | Euroscarf |
| SMy306 | atg16::kan; ATG5-K57E,N84E-TAP-cyc1term-URA-tTEF | BY474x | this study |
| SMy308 | atg16::kan; ATG5-TAP-cyc1term-URA-tTEF | BY474x | this study |
| SMy342 | Mat a; ATG5-TAP:URA3, atg19::KanMX, atg16::KanMX | BY474x | this study |
| SMy344 | Mat a; ATG5-K57E,N84E-TAP:URA3, atg19::KanMX, atg16::KanMX | BY474x | this study |
| SMy346 | atg19::KanMX, atg16::KanMX | BY474x | this study |
| SMy356 | atg16::kan; pho13::kan; pho8△60::His; ATG5-TAP-cyc1term-URA-tTEF | BY474x | this study |
| SMy358 | atg16::kan; pho13::kan; pho8△60::His; ATG5-K57E,N84E-TAP-cyc1term-URA-tTEF | BY474x | this study |
| yAB7 | atg13::KanMX, ATG17-TEV-2xProtA-4xH3-5xHA:URA | BY474x | (*Brezovich et al., 2015*) |

Quantification of microscopy GFP-TRAP experiments (*Figure 1G*) was performed using ImageJ Software. A z-projection using the maximum intensity values was generated from every stack. Intensity values of 30 pixels along the rim of each bead (oval selected in the GFP-channel but values measured in the mCherry-channel) were measured with Oval_Profile.java plugin, averaged and the background from a representative empty area in the image was subtracted.

For experiments shown in *Figures 1H* and 5 µL of GFP-TRAP beads (Chromotek, Germany) were mixed with 15 µL of empty sepharose 4B beads per each sample, and washed three times in IP wash buffer (20 mM Tris pH8, 135 mM NaCl, 10% glycerol). 15 µL of lysates were taken as input (approx. 7.5% of the total volume). The remaining lysate was added to the beads and incubated for 1 hr rotating at 4°C. Beads were washed three times with IP wash buffer and finally resuspended in 15 µL SDS loading dye. Input and bead samples were subjected to Western blot analysis. Membranes were ultimately developed with Clarity ECL substrate (BioRad) or with SuperSignal West Femto substrate (ThermoFisher Scientific, Rockford, IL, USA) if needed. The signal was recorded with a Chemi-DOC Touch (Biorad) imager. For quantification of band intensities, only non-saturated exposures were considered, lanes were defined in ImageJ and the lane profile plotted. The area under the peak of relevant bands was taken as readout. The beads/input enrichment factor (EF) was calculated according to the following equation:

$$\frac{BEADS_{GFP-ATG5}/BEADS_{GFP}}{INPUT_{GFP-ATG5}/INPUT_{GFP}} * \frac{GAPDH_{GFP-ATG5}}{GAPDH_{GFP}}$$

Where *BEADS* and *INPUT* indicate the cargo receptor's band intensity in the respective fraction, *GAPDH* indicates the band intensity of the anti-GAPDH blot, and *GFP-ATG5* and *GFP* indexes denote the sample. Interactions were considered reliable only for EF > 2.

## Antibodies

The following antibodies were used for detection of proteins in GFP-TRAP experiments on HeLa lysates are. Mouse anti-p62 (BD Biosciences, #610832, Franklin Lakes, NJ, USA) was used at 1:1000 dilution; Rabbit anti-NDP52 (Cell Signaling, #9036) was used at 1:1000 dilution; Rabbit anti-OPTN (Sigma, HPA003279) was used at 1:500 dilution; Mouse anti-GFP (Roche, cat. 11 814 460 001) was used at 1:1000 dilution; Mouse anti-GAPDH (Sigma, Clone GAPDH-71.1) was used at 1:25000 dilution. HRP-conjugated goat anti-Rabbit and anti-Mouse (Jackson

ImmunoResearch, #111-035-003 and 115-035-003, respectively) were used at 1:10000 dilution.

## Yeast strains

All strains of *S.cerevisiae* S288C BY474x genetic background are derived from the diploid strain BY4743 and carry the following markers: his3Δ1; leu2Δ0; met15Δ0; ura3Δ0 except if stated otherwise. ATG5-TAP, ATG5-K57E,N84E-TAP, ATG16-TEV-2xProtA-4xH3-5xHA, ATG17-TEV-2xProtA-4xH3-5xHA strains were generated by homologous recombination of the tagged protein into the respective deletion strains. All other strains were generated by crossing of single strains.

## Molecular dynamics simulations

The initial coordinates of Atg5 (with a part of Atg16 bound to it) were obtained from the protein data bank (PDB) with identifier 2DYO (*Matsushita et al., 2007*). The missing loops and atoms were modeled using the SWISS-MODEL server (*Arnold et al., 2006*; *Biasini et al., 2014*; *Bordoli et al., 2009*). The final model contained residues 1–285 of Atg5 and 22–57 of Atg16 (numbered according to 2DYO).

The protonation states of the histidine residues were determined with the WHATIF server (*Vriend, 1990*). In order to capture the flexibility of the protein, molecular dynamics simulations were used to generate an ensemble of protein structures. These simulations, as well as the preparatory energy minimizations, were performed using GROMOS11 (*Schmid et al., 2012*) in combination with the Gromos force field 54a8 (*Reif et al., 2012*). Initially, the Atg5-Atg16 complex was minimized in vacuum using steepest descent for 2000 steps. Subsequently, the complex was solvated in a rectangular box with 22,250 SPC water molecules (*Berendsen et al., 1981*). The overall system was electrostatically neutral and therefore no ions were added. The solvent configurations were relaxed with another round of energy minimization where the solute atoms were position-restrained.

Initial random velocities were drawn from a Maxwell-Boltzmann distribution at 50 K. Position restraints on the solute atoms were applied with an initial force constant of $2.5 \times 10^4$ kJ mol$^{-1}$ nm$^{-2}$. With each step of equilibration performed at constant volume, the temperature was increased by 50 K, the force constant of the position restraints was reduced by a factor of 10 and the system was simulated for 20 ps. The final equilibration step was simulated for 40 ps at 298 K, without any position restraints. After these equilibration steps, the system was simulated for 1 ns at a constant temperature of 298 K using weak coupling at constant volume (*Berendsen et al., 1984*). Two separate temperature baths were used for the solute and solvent and the relaxation time was set to 0.1 ps. All bond lengths were constrained using the SHAKE algorithm (*Ryckaert et al., 1977*) with a geometric accuracy of $1 \times 10^{-4}$, which enabled the use of a two fs time step. The center of mass translation motion was removed every 1000 steps. A triple range cut-off scheme was used to calculate the non-bonded interactions and a pair-list was generated every fifth time step. Interactions within 0.8 nm were calculated at every time step, whereas the interactions between 0.8 and 1.4 nm were evaluated only when the pair-list was updated and kept constant at intermediate time steps. The interactions beyond 1.4 nm were approximated by a reaction field contribution, representing a homogeneous medium with a dielectric constant of 61, as appropriate for SPC water molecules (*Heinz et al., 2001*).

## Docking

The molecular dynamics simulation, as described above, was used to obtain different configurations of the Atg5-Atg16 complex that could be used for docking. The initial configuration of Atg5-Atg16, as well as ten snapshots obtained from the simulation (sampled every 100 ps) were used to take the flexibility of the proteins into account during the subsequent docking procedure. The C-terminal peptide TWEEL of Atg19 was modeled with NH and COO$^-$ termini. The NH terminus was chosen to represent the NH group of the peptide bond that would be present in the complete Atg19. Auto-Dock Vina (*Trott and Olson, 2010*) was used to dock the peptide into the configurations of the Atg5-Atg16 complex. The exhaustiveness was set to 50 and the search space was defined such that the whole complex was searched. The peptide was completely flexible during the docking process, whereas the Atg5-Atg16 complex was kept rigid. For each of the configurations of Atg5-Atg16, the nine best poses of TWEEL were evaluated. The docking results were manually examined in order to discard any poses in which the Thr amino acid of the peptide was completely buried. These poses would not be possible with the complete Atg19 and are therefore not of interest. Three major interaction sites were found to be reoccurring in multiple snapshots of the Atg5-Atg16 complex. One of them involved residues of both Atg5 and Atg16. Since the focus of the present study was identification of the interactions of TWEEL with Atg5, this interaction site was no longer considered. For other binding sites, the hydrogen bonding patterns were examined for all the poses of the docked peptide in all of the configurations of Atg5-Atg16. Generally, several configurations of the peptide were found in each of the binding sites, but here we focused on the residues that were most prone to be involved in salt bridges or hydrogen bonds to the TWEEL peptide. The first binding site, which was also found in the initial configuration of the Atg5-Atg16 complex, was characterized by persistent salt bridges and hydrogen bonds between the glutamic acids of the peptide with K57 and K137. In the second binding site, the residues N84 and R208 were the ones that were most often involved hydrogen bonds and salt bridges with TWEEL.

## Superposition of protein structures

The Atg8 (PDB: 2ZPN, (*Noda et al., 2008*) and Atg5 (PDB: 2DYO, (*Matsushita et al., 2007*) structures were superposed by secondary-structure matching (SSM) (*Krissinel and Henrick, 2004*) using the Coot software (*Emsley and Cowtan, 2004*). The Atg8 molecule in complex with Atg19 AIM motif was superposed onto the two ubiquitin-like Atg5 bundles separately. The resulting shift of Atg19 AIM motif was depicted omitting the original Atg8 structure for clarity, indicating putative AIM binding pockets on Atg5.

## Acknowledgements

The authors are supported by ERC grants (No. 260304, No. 646653 to SM and No. 279408 to Bojan Zagrovic), by Austrian Science Foundation (FWF) grants (No. P25546-B20 to SM; No. P 25522-B20

and P 28113-B28 to CK; No. T724-B20 to JSM), by a Vienna Research Groups for Young Investigators grant from the Vienna Science and Technology Fund (WWTF, VRG10-791 001 to CK) and by the EMBO Young Investigator Program to SM and CK. We thank Anete Romanauska for assistance with expression, purification and testing Atg5 mutant proteins in pull down experiments and Adriana Savova for assistance in the p62 interaction experiments. We thank Nina Hobl for cloning, expressing and purifying Atg5-mCherry.

## Additional information

### Funding

| Funder | Grant reference number | Author |
|---|---|---|
| Austrian Science Fund | T724-B20 | Justyna Sawa-Makarska |
| Austrian Science Fund | P25522-B20 | Claudine Kraft |
| Austrian Science Fund | P28113-B28 | Claudine Kraft |
| Vienna Science and Technology Fund | VRG10-791 001 | Claudine Kraft |
| European Molecular Biology Organization | YIP | Claudine Kraft Sascha Martens |
| European Research Council | 279408 | Bojan Zagrovic |
| European Research Council | 260304 | Sascha Martens |
| European Research Council | 646653 | Sascha Martens |
| Austrian Science Fund | P25546-B20 | Sascha Martens |

The funders had no role in study design, data collection and interpretation, or the decision to submit the work for publication.

### Author contributions

DF, Conception and design, Acquisition of data, Analysis and interpretation of data, Drafting or revising the article, Contributed unpublished essential data or reagents; JS-M, Conception and design, Acquisition of data, Analysis and interpretation of data, Drafting or revising the article, Contributed unpublished essential data or reagents ; BZe, Acquisition of data, Analysis and interpretation of data, Contributed unpublished essential data or reagents ; AdR, Conception and design, Acquisition of data, Analysis and interpretation of data, Contributed unpublished essential data or reagents; GZ, Acquisition of data, Analysis and interpretation of data; AB, Acquisition of data, Contributed unpublished essential data or reagents ; JR, KR, Acquisition of data, Analysis and interpretation of data, Contributed unpublished essential data or reagents; CK, Conception and design, Analysis and interpretation of data, Contributed unpublished essential data or reagents; BZa, Conception and design, Analysis and interpretation of data, Drafting or revising the article; SM, Conception and design, Analysis and interpretation of data, Drafting or revising the article, Contributed unpublished essential data or reagents

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
