## [Decision Letter]

Thank you for submitting your article "Mechanism of cargo-directed Atg8 conjugation during selective autophagy" for consideration by *eLife*. Your article has been reviewed by three peer reviewers, one of whom is a member of our Board of Reviewing Editors, and the evaluation has been overseen by Ivan Dikic as the Senior Editor. The reviewers have opted to remain anonymous.

The reviewers have discussed the reviews with one another and the Reviewing Editor has drafted this decision to help you prepare a revised submission.

Summary:

The reviewers agreed that the authors present a mechanistic study describing cargo-stimulated Atg8 lipidation via Atg5/12/16 in biochemical reconstitution studies, providing a novel insight into the mechanism of autophagosome formation during selective autophagy. However, additional work is required to strengthen the manuscript to meet the standards for *eLife*. There was a general sense that much more clarity and caution needs to be applied to the description of experiments, results and, most importantly, conclusions.

Essential revisions:

1) Bulk autophagy (starvation-induced non-selective autophagy) should be examined in the Atg5 mutant cells.

2) The binding assays with the Atg5 mutants are largely in-vitro with purified components on beads. The interactions of these mutants in-vivo should be tested either by co-IP or perhaps some other method like proximity labeling, given that the effects on prApe1 processing are modest at best.

3) Additional assays or replicates are necessary to demonstrate that the mapped residues in Atg5 (N84E, K57/N84E) are the key mediators of the Atg5-Atg19 interaction. Reviewers have a concern that reduced prApe1 processing in the Atg5 mutant strains presented in Figure 5 could be attributed to reduced loading of the gel (with quite large error across the replicate experiments) and also the in vitro lipidation assay in Figure 5 shows a minimal defect in Atg8 lipidation in the Atg5 double mutant.

4) Several other adaptors such as p62 should be examined to investigate their interactions with Atg5-12-16 to demonstrate that the mechanism is not limited to yeast and has a broad impact.

[Editors' note: further revisions were requested prior to acceptance, as described below.]

Thank you for resubmitting your work entitled "Mechanism of cargo-directed Atg8 conjugation during selective autophagy" for further consideration at *eLife*. Your revised article has been favorably evaluated by Ivan Dikic (Senior editor), a Reviewing editor, and one reviewer.

The manuscript has been improved but there are some remaining issues that need to be addressed before acceptance, as outlined below:

1) The authors should describe how the E102A mutation affects the function of Atg16. On the other hand, I also felt that newly added data using Atg16 mutants are less informative and thus can be omitted from the final version of the manuscript or transferred to figure supplements.

2) Figure 1: Since in the experiment using GST alone, a comparable amount of GST was not bound to the beads, this cannot serve as a control.

3) Figure 1: It is unclear whether the control strain (shown in the left lane) expressed 9xmyc-SUV alone or not. A strain expressing it should be examined.

4) Figure 1: A blot for anti-GFP should also be shown.

5) Figure 2: Images for GST proteins should also be added.

6) Figure 3: The label 407 is put at the wrong position.

---

## [Author Response]

**[…]**

*Essential revisions:*

*1) Bulk autophagy (starvation-induced non-selective autophagy) should be examined in the Atg5 mutant cells.*

We have now characterized the Atg5 mutants much more thoroughly. In order to test the effect of the K57E,N84E mutations in Atg5 on starvation-induced bulk autophagy we conducted the GFP-Atg8 cleavage assay. The three replicates performed by us show no impairment of bulk autophagy in cells expressing the K57E,N84E mutant. Please see Figure 5 for a representative blot.

We also performed Pho8Δ60 activity assays in order to test the effect of the K57E, N84E and K57E, N84E mutations on bulk autophagy. The experiments performed show no difference in activity of the Pho8Δ60 enzyme between wild type Atg5 and the mutants. Please refer to Figure 5 for quantifications. Thus, our data strongly suggest that the K57E, N84E mutant does not affect bulk autophagy.

*2) The binding assays with the Atg5 mutants are largely in-vitro with purified components on beads. The interactions of these mutants in-vivo should be tested either by co-IP or perhaps some other method like proximity labeling, given that the effects on prApe1 processing are modest at best.*

We have conducted co-IP experiments with Atg5 K57E, N84E and a representative blot of such an experiment is shown in Figure 5. In particular, we found that the K57E,N84E mutation resulted in a robust decrease in the interaction with Atg19. In order to corroborate this result, we went a step further and investigated the Atg5 – Atg19 interaction in context of the Atg16 E102A mutation. This mutation is based on a previous finding by Juris et al., 2015, EMBOJ (PMID: 25691244) reporting that Atg21 recruits Atg16 to the PAS during the Cvt pathway and that the E102A mutation in Atg16 impairs the Atg16 – Atg21 interaction. We reasoned that this interaction may indirectly stabilize the Atg12~Atg5-Atg16 – Atg19 complex. Indeed, we found an even further decrease of the Atg12~Atg5-Atg16 – Atg19 interaction in context of the Atg16 E102A mutation (Figure 5).

*3) Additional assays or replicates are necessary to demonstrate that the mapped residues in Atg5 (N84E, K57/N84E) are the key mediators of the Atg5-Atg19 interaction. Reviewers have a concern that reduced prApe1 processing in the Atg5 mutant strains presented in Figure 5 could be attributed to reduced loading of the gel (with quite large error across the replicate experiments) and also the* in vitro *lipidation assay in Figure 5 shows a minimal defect in Atg8 lipidation in the Atg5 double mutant.*

We have replaced the blot shown in Figure 5 with a technically better blot (see Figure 5). We also conducted additional prApe1 processing assays that confirm the impairment of prApe1 processing by the Atg5 K57E, N84E mutation (Figure 5 and Figure 5—figure supplement 2). This included the analysis of the prApe1 transport in context of the Atg16 D101A, E102E mutation, which was reported to reduce the recruitment of Atg12~Atg5-Atg16 by Atg21 to the PAS (please see Juris et al., 2015, EMBOJ (PMID: 25691244) and our reply to your comment number 2).

Moreover, we have conducted additional in vitro lipidation assays and quantified them. The graph of this quantification has been inserted in Figure 5 and it shows that the Atg5 mutants do not display significant defects in promoting Atg8~PE conjugation. A more representative gel of the in vitro conjugation assay has been inserted in Figure 5—figure supplement 1 in place of the previous one (right panel).

*4) Several other adaptors such as p62 should be examined to investigate their interactions with Atg5-12-16 to demonstrate that the mechanism is not limited to yeast and has a broad impact.*

In our previous version we could already demonstrate that the human ATG5 coprecipitates p62. Upon request of the reviewers we have now extended this analysis and demonstrate that also the Optineurin and NDP52 cargo receptors co-precipitate with ATG5 (please see Figure 1). Thus the interaction of cargo receptors with components of the Atg12~Atg5-Atg16 appears to be a more general phenomenon. At this point we do not know if this interaction is direct and in order to thoroughly investigate this interaction we would have to have the recombinant human Atg12~Atg5-Atg16 complex in our hands, which is currently not the case.

[Editors' note: further revisions were requested prior to acceptance, as described below.]

*The manuscript has been improved but there are some remaining issues that need to be addressed before acceptance, as outlined below:*

*1) The authors should describe how the E102A mutation affects the function of Atg16. On the other hand, I also felt that newly added data using Atg16 mutants are less informative and thus can be omitted from the final version of the manuscript or transferred to figure supplements.*

Thank you for raising this point. We have now moved the main experiments showing data obtained with the Atg16 mutant to the figure supplements (Figure 5—figure supplement 3 and Figure 5—figure supplement 4). We have decided to keep the IP experiments including the Atg16 E102A mutant in the main figure (Figure 5) since it corroborates the interaction of Atg5 with Atg19 in vivo and suggests that there may be an additional recruitment of the Atg12~Atg5-Atg16 to the PAS by Atg21.

*2) Figure 1: Since in the experiment using GST alone, a comparable amount of GST was not bound to the beads, this cannot serve as a control.*

Thank you for raising this point. We have now moved the main experiments showing data obtained with the Atg16 mutant to the figure supplements (Figure 5—figure supplement 3 and Figure 5—figure supplement 4). We have decided to keep the IP experiments including the Atg16 E102A mutant in the main figure (Figure 5) since it corroborates the interaction of Atg5 with Atg19 in vivo and suggests that there may be an additional recruitment of the Atg12~Atg5-Atg16 to the PAS by Atg21.

*3) Figure 1: It is unclear whether the control strain (shown in the left lane) expressed 9xmyc-SUV alone or not. A strain expressing it should be examined.*

The control you are referring to was conducted in the study that originally reported the technique (Brezovich, Schuschnig, Ammerer, Kraft, 2015, Yeast). In particular, Figure 2 of this study shows that overexpression of a myc-tagged methyltransferase from a Gal promoter does not result in unspecific methylation of the H3 tail. Since the methyltransferase fused to Atg19 is expressed from the Atg19 promoter and thus at much lower levels compared to Gal promoter driven expression, we have not included this control in our experiments. We have added an explanatory sentence to the text (Results section).

*4) Figure 1: A blot for anti-GFP should also be shown.*

We have now added anti-GFP blots for this experiment to Figure 1—figure supplement 1.

*5) Figure 2: Images for GST proteins should also be added.*

We have now added images of the GST proteins to Figure 2—figure supplement 2.

*6) Figure 3: The label 407 is put at the wrong position.*

Thank you for pointing this out. This mistake has been corrected.